# FreDF: Learning to Forecast in the Frequency Domain

**Hao Wang**[1]    **Licheng Pan**[1]    **Yuan Shen**[1]    **Zhichao Chen**[1]    **Degui Yang**[2]
**Yifei Yang**[3]    **Sen Zhang**[4]    **Xinggao Liu**[1]    **Haoxuan Li**[5*]    **Dacheng Tao**[6*]

[1]Department of Control Science and Engineering, Zhejiang University
[2]School of Automation, Central South University
[3]Department of Computer Science and Engineering, Shanghai Jiao Tong University
[4]Trust and Safety Team, TikTok Sydney, ByteDance Inc.
[5]Center for Data Science, Peking University
[6]Generative AI Lab, College of Computing and Data Science, Nanyang Technological University
`Ho-ward@outlook.com`  `hxli@stu.pku.edu.cn`  `dacheng.tao@ntu.edu.sg`

## Abstract

Time series modeling presents unique challenges due to autocorrelation in both historical data and future sequences. While current research predominantly addresses autocorrelation within historical data, the correlations among future labels are often overlooked. Specifically, modern forecasting models primarily adhere to the Direct Forecast (DF) paradigm, generating multi-step forecasts independently and disregarding label autocorrelation over time. In this work, we demonstrate that the learning objective of DF is biased in the presence of label autocorrelation. To address this issue, we propose the Frequency-enhanced Direct Forecast (FreDF), which mitigates label autocorrelation by learning to forecast in the frequency domain, thereby reducing estimation bias. Our experiments show that FreDF significantly outperforms existing state-of-the-art methods and is compatible with a variety of forecast models. Code is available at `https://github.com/Master-PLC/FreDF`.

## 1 Introduction

Time series modeling aims to utilize historical sequence to predict future data, which has been successfully applied in various fields (Qiu et al., 2025), including long-term forecasting in weather prediction (Bi et al., 2023), short-term predictions in security (Yan et al., 2024), and data imputation in industrial maintenance (Wang et al., 2024b). A key challenge in time series modeling, distinguishing it from canonical regression tasks, is the presence of autocorrelation, which refers to the dependence between time steps inherent in *both* the input and label sequences.

To accommodate autocorrelation in input sequences, diverse forecast models have been developed, exemplified by recurrent (Salinas et al., 2020), convolution (Wu et al., 2023) and graph neural networks (Yi et al., 2023a). Recently, Transformer-based models, utilizing self-attention mechanisms to dynamically assess autocorrelation, have gained prominence (Liu et al., 2024; Nie et al., 2023). Concurrently, there is a growing trend of incorporating frequency analysis into forecast models. By representing the input sequence in the frequency domain, input autocorrelation can be efficiently accommodated, which proves to improve the forecast performance of Transformers (Zhou et al., 2022) and Multi-Layer Perceptrons (MLPs) (Yi et al., 2023b). These pioneering works highlight the importance of autocorrelation and frequency analysis in advanced time series modeling.

Another critical aspect is the autocorrelation within the label sequence, where each future step is autoregressively dependent on its predecessors. This phenomenon, termed as *label autocorrelation*, poses a critical issue warranting investigation. Specifically, recent forecasting methods predominantly employ the Direct Forecast (DF) paradigm (Liu et al., 2024; Nie et al., 2023), which generates multi-step predictions simultaneously via a multi-output head (Liu et al., 2022b), optimizing forecast errors across all steps concurrently. However, this approach implicitly assumes *step-wise independence* in

---

*Corresponding author.

the label sequence, overlooking the label autocorrelation inherent in the time series forecast task. We theoretically demonstrate that this oversight results in biased forecasts, revealing a significant defect with the existing DF paradigm.

To address this issue, we introduce the *Frequency-enhanced Direct Forecast* (FreDF), a straightforward yet effective refinement of the DF paradigm. The central idea is to align the forecasts and label sequences in the frequency domain, where the label correlation is found to be effectively diminished. This method resolves the discrepancy between the scope of DF and the characteristics of actual time series, while retaining DF's advantages, such as sample efficiency and simplicity of implementation. Our main contributions are summarized as follows:

- We uncover label autocorrelation as a critical yet underexplored challenge in modern time series modeling and theoretically justify how it biases the learning objective of the prevalent DF paradigm.
- We propose FreDF, a straightforward yet effective modification to the DF paradigm that learns to forecast in the frequency domain, thereby mitigating label autocorrelation and reducing bias. To our knowledge, this is the first effort to utilize frequency analysis for enhancing forecast paradigms.
- We validate the efficacy of FreDF through comprehensive experiments, demonstrating its ability to enhance the performance of state-of-the-art forecasting models across a diverse range of datasets.

## 2 PRELIMINARIES AND RELATED WORK

### 2.1 PROBLEM DEFINITION

In this study, uppercase letters (e.g., $Y$) denote random matrix, with subscripts (e.g., $Y_{i,j}$) indicating matrix entries. An uppercase letter followed by parentheses (e.g., $Y(n)$) represents an observation of the random matrix. A multi-variate time series can be represented as a sequence $[X(1), X(2), \cdots, X(\mathrm{N})]$, where $X(n) \in \mathbb{R}^{1 \times \mathrm{D}}$ is the sample at the $n$-th timestamp with D covariates. Define input sequence $L \in \mathbb{R}^{\mathrm{H}} \times \mathrm{D}$ and label sequence $Y \in \mathbb{R}^{\mathrm{T} \times \mathrm{D}}$ where H and T are sequence lengths. At an arbitrary $n$-th step, these sequences are observed as $L = [X(n - \mathrm{H} + 1), ..., X(n)]$ and $Y = [X(n + 1), ..., X(n + \mathrm{T})]$. The goal of time series forecast is identifying a model $g : \mathbb{R}^{\mathrm{H} \times \mathrm{D}} \to \mathbb{R}^{\mathrm{T} \times \mathrm{D}}$ within a model family $\mathcal{G}$ (e.g., decision trees, neural networks) that generates the prediction sequence $\hat{Y} = g(L)$ approximating the label sequence $Y$.

There are two critical aspects to accommodate autocorrelation in time series modeling: (1) selecting a model family $\mathcal{G}$ that encodes autocorrelation in input sequences, which underscores the design of model architectures; (2) generating forecasts that respect label autocorrelation, which highlights the efficacy of forecast paradigms. Our survey concentrates on examining both aspects.

### 2.2 MODEL ARCHITECTURES

To exploit autocorrelation in the input sequences, a variety of architectures have been developed (Qiu et al., 2024; Li et al., 2024c). Initial statistical methods include VAR (Watson, 1993) and ARIMA (Asteriou & Hall, 2011). Subsequently, neural networks gained prominence for their ability to automate feature interaction and capture nonlinear correlations. Exemplars include RNNs (e.g., DeepAR (Salinas et al., 2020), S4 (Gu et al., 2021)), CNNs (e.g., TimesNet (Wu et al., 2023)), and GNNs (e.g., MTGNN (Mateos et al., 2019)), each designed to effectively encode autocorrelation. Current progress has reached a debate between Transformer-based and MLP-based architectures, each with its advantages and limitations. Transformers (e.g., PatchTST (Nie et al., 2023), iTransformer (Liu et al., 2024)) offer significant scalability as data size increases but incur high computational costs; MLPs (e.g., DLinear (Zeng et al., 2023), TimeMixer (Wang et al., 2024c)) are generally more efficient but less effective in scaling with larger datasets and struggle to accommodate varying input lengths.

An emerging approach is representing sequence in the frequency domain (Wu et al., 2021; 2025). This method, in comparison to modeling autocorrelation in the temporal domain, manages autocorrelation effectively with limited cost. A prominent example is FedFormer (Zhou et al., 2022), which computes attention scores in the frequency domain, leading to improved efficiency, efficacy, and noise reduction capabilities. The success of this technique extends to various architectures like Transformers (Zhou et al., 2022; Wu et al., 2021), MLPs (Yi et al., 2023b) and GNNs (Yi et al., 2023a; Cao et al., 2020), which makes it a versatile plugin in the design of neural networks for time series forecast.

## 2.3 ITERATIVE FORECAST V.S. DIRECT FORECAST

There are two paradigms to generate multi-step forecast: iterative forecast (IF) and direct forecast (DF) (Liu et al., 2022b). *The IF paradigm follows the canonical sequence-to-sequence manner*, which forecasts one step at a time and uses previous predictions as input for subsequent forecasts. This recursive approach respects label autocorrelation in forecast generation, widely used by early-stage methods (Lai et al., 2018; Salinas et al., 2020). However, IF suffers from high variance due to error propagation, which significantly impairs performance in long-term forecasts (Taieb & Atiya, 2015). Therefore, modern works (Li et al., 2021) advocate the DF paradigm, which generates multi-step forecasts simultaneously using a multi-output head, featured by fast inference, implementation ease and superior accuracy. Currently, DF has been a dominant paradigm, continuing to be employed in modern works (Wu et al., 2023; Liu et al., 2024).

**Significance of this work.** Our work refines the DF paradigm by performing forecasting in the frequency domain[1]. In contrast to recent advancements that incorporate frequency analysis within model architectures to manage *input autocorrelation* (Yi et al., 2023a;b; Wang et al., 2025), accelerate computation (Lange et al., 2021), and improve generation quality (Yuan & Qiao, 2024), our approach specifically focuses on refining the loss function to mitigate the bias caused by *label autocorrelation*, which is an unexplored yet significant aspect in modern time series analytics.

## 3 PROPOSED METHOD

### 3.1 MOTIVATION

Autocorrelation is a fundamental characteristic of time series data, where each observation is highly dependent on previous ones (Zeng et al., 2023). This characteristic sets time series apart from other types of data and creates specific modeling challenges. To accommodate autocorrelation, various neural network architectures have been developed (Wu et al., 2021; Liu et al., 2024), which effectively model the autocorrelation in input sequence. However, label autocorrelation cannot be handled via the modification of neural architectures. To effectively manage label autocorrelation, it is necessary to create learning objectives that specifically consider these dependencies.

Modern time series forecasting models are primarily trained under the multitask learning manner, known as the direct forecasting (DF) paradigm. Specifically, the DF paradigm employs a multi-output model $g_\theta : \mathbb{R}^{H \times D} \to \mathbb{R}^{T \times D}$ to generate $T$-step forecasts $\hat{Y} = g_\theta(L)$. The model parameters $\theta$ are optimized by minimizing the temporal loss:

$$\mathcal{L}^{(\mathrm{tmp})} := \sum_{t=1}^{T} \left\| Y_t - \hat{Y}_t \right\|_2^2. \tag{1}$$

In this learning objective, the temporal loss at each forecast step is computed independently, treating each future time step as a separate task. While this method has shown empirical effectiveness, it overlooks the autocorrelation present within the label sequence $Y$. Specifically, the label sequence is autoregressively generated, with $Y_{t+1}$ being highly dependent on $Y_t$, as illustrated by the blue arrows in Fig. 1(a). In contrast, the learning objective in (1) assumes that each step in the label sequence can be independently modeled, as indicated by the black arrows in Fig. 1(a). This misalignment between the model's assumptions and the data's characteristics introduces bias into the learning objective of the DF paradigm, as demonstrated in Theorem 3.1.

**Theorem 3.1** (Bias of DF). *Given input sequence $L$ and label sequence $Y$, the learning objective* (1) *of the DF paradigm is biased against the practical negative-log-likelihood (NLL), expressed as:*

$$\text{Bias} = \sum_{i=1}^{T} \frac{1}{2\sigma^2} (Y_i - \hat{Y}_i)^2 - \sum_{i=1}^{T} \frac{1}{2\sigma^2(1-\rho_i^2)} \left( Y_i - \left( \hat{Y}_i + \sum_{j=1}^{i-1} \rho_{ij}(Y_j - \hat{Y}_j) \right) \right)^2, \tag{2}$$

*where $\hat{Y}_i$ indicates the prediction at the $i$-th step, $\rho_{ij}$ denotes the partial correlation between $Y_i$ and $Y_j$ given $L$, $\rho_i^2 = \sum_{j=1}^{i-1} \rho_{ij}^2$.*

---

[1]Given the inferior performance of the IF paradigm (Li et al., 2021), this paper advocates adapting the DF paradigm to handle label autocorrelation, rather than revisiting IF to directly model label autocorrelation.

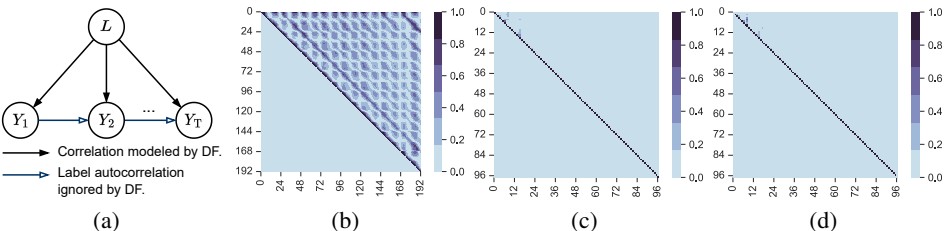

(a)  (b)  (c)  (d)

Figure 1: Visualizing label autocorrelation in time series forecasting. (a) shows the generation process of time series with dependencies depicted as arrows. (b) shows the label correlation in the time domain, where each element $\rho_{i,j}$ indicates the partial correlation between $Y_i$ and $Y_j$ given $L$. (c-d) shows the label correlation in the frequency domain, where each element $\rho_{i,j}$ indicates the partial correlation between $F_i$ and $F_j$ given $L$, shown with the real (c) and imaginary part (d). Due to the symmetry inherent in FFT, the forecast length in the frequency domain is halved.

According to Theorem 3.1, the presence of label autocorrelation $\rho_{ij}$ causes the loss to be biased against the NLL of the real data. Notably, this bias diminishes to zero when the labels are uncorrelated ($\rho_{ij} = 0$). Therefore, label autocorrelation is a crucial aspect for training time series forecast models.

## 3.2 REDUCE LABEL AUTOCORRELATION WITH FOURIER TRANSFORM

As established in Theorem 3.1, the bias in the learning objective decreases as label autocorrelation diminishes. To achieve this reduction, a promising strategy is transforming the label sequence into a representation where autocorrelation is minimized. The Discrete Fourier Transform (DFT), defined in Definition 3.2, offers an intuitive approach, which projects the sequence onto a set of orthogonal exponential bases. In this transformed space, the label sequence is described as a linear combination of predefined temporal patterns that are orthogonal, which effectively bypasses the autocorrelation in the time domain. The efficacy of this transformation in reducing label autocorrelation is formalized in Theorem 3.3, where different frequency components become decorrelated. Consequently, the reduced $\rho_{i \neq j}$ lowers the bias against the NLL, which benefits the training of time series forecast models.

**Definition 3.2** (Discrete Fourier Transform, DFT). *The normalized DFT of a sequence $Y = [Y_0, ..., Y_{T-1}]$ is defined as the projection onto a set of orthogonal Fourier bases at different frequencies. The projection for frequency $k$ is computed as*

$$F_k = \sum_{t=0}^{T-1} Y_t \exp\left(-j(\frac{2\pi k}{T})t\right) / \sqrt{T},$$

*where $j$ is the imaginary unit , $\exp(\cdot)$ is the Fourier basis for different $k$ values. The DFT comprises the set of projections $F = [F_1, ..., F_{T-1}]$, denoted as $F = \mathcal{F}(Y)$, which can be computed via the Fast Fourier Transform (FFT) algorithm with complexity $\mathcal{O}(T \log T)$.*

**Theorem 3.3** (Decorrelation between frequency components). *Let $Y$ be a zero-mean, discrete-time, wide-sense stationary random process of length $T$. As $T \to \infty$, the DFT coefficients become asymptotically uncorrelated at different frequencies:*

$$\lim_{T \to \infty} \mathbb{E}[F_k F_{k'}^*] = \begin{cases} S_Y(f_k), & \text{if } k = k', \\ 0, & \text{if } k \neq k', \end{cases}$$

*where $f_k = \frac{k}{T}$ and $S_Y(f)$ is the power spectral density of $Y$ .*

**Case study.** To validate our theoretical claims, we conducted a case study on the Weather dataset, illustrated in Fig. 1.Implementation details and additional evidence are provided in Appendix A. The main observations are summarized as follows:

- **Evidence of Label Autocorrelation:** Fig. 1 (b) quantifies the partial correlations between different steps $Y_i$ and $Y_j$ of the label sequence $Y$, conditioned on the input $L$. A number of non-diagonal

elements exhibit substantial values, with approximately 37.5% exceeding 0.3. This indicates that different time steps in $Y$ are correlated conditioned on $L$, confirming the presence of label autocorrelation. Moreover, the autocorrelation displays regular variations, evidenced by alternating light and dark regions in Fig. 1 (b), suggesting a periodic nature in the series. Such label autocorrelation makes the learning objective of the naive DF paradigm biased, as established in Theorem 3.1.

- **Effect of Domain Transformation:** Fig. 1 (c-d) visualize the partial correlations between different frequency components of the transformed label sequence $F$. The majority of non-diagonal elements show negligible values, with only about 3.6% exceeding 0.1. This demonstrates that transforming the label sequence to the frequency domain significantly reduces the partial correlations between different components, corroborating Theorem 3.3. The reduction in label correlation $\rho_{i \neq j}$ leads to a decrease in the bias identified in Theorem 3.1, underscoring the potential of forecasting in the frequency domain for more accurate and unbiased predictions.

### 3.3 Model Implementation

This section introduces FreDF, an innovative approach that enhances the vanilla Direct Forecast (DF) training paradigm. FreDF aligns forecast and label sequences within the frequency domain, effectively mitigating the bias introduced by label autocorrelation.

As illustrated in Fig. 2, the input sequence $L$ is fed into the model to generate $T$-step forecasts, expressed as $\hat{Y} = g(L)$. The temporal forecast error $\mathcal{L}^{(\text{tmp})}$ is computed according to (1). Subsequently, both the forecast and label sequences are transformed into the frequency domain using FFT. The frequency forecast error is then calculated as:

$$\mathcal{L}^{(\text{feq})} := \left| \mathcal{F}(\hat{Y}) - \mathcal{F}(Y) \right|_1, \quad (3)$$

Figure 2: The workflow of FreDF. Key operations in the time and frequency domains are highlighted in red and blue, respectively.

where $Y \in \mathbb{R}^{T \times D}$, $|\cdot|_1$ denotes the element-wise $\ell_1$ norm, summing the absolute values of all elements within the matrix. Since FFT is differentiable (Wu et al., 2021; Zhou et al., 2022), $\mathcal{L}^{(\text{freq})}$ can be optimized using standard stochastic gradient descent methods. We advocate the use of the $\ell_1$ loss in the frequency domain instead of the squared loss due to the numerical characteristics of the transformed label sequence. Specifically, different frequency components often exhibit vastly varying magnitudes; lower frequencies possess significantly higher amplitudes compared to higher frequencies, making the squared loss prone to instability. By using the $\ell_1$ loss, we seek for a more balanced and stable optimization process.

Finally, the temporal and frequency forecast errors are fused, with the weighting parameter $0 \leq \alpha \leq 1$ controlling the relative contribution of each error:

$$\mathcal{L}^{\alpha} := \alpha \cdot \mathcal{L}^{(\text{feq})} + (1 - \alpha) \cdot \mathcal{L}^{(\text{tmp})}. \quad (4)$$

By aligning the forecast and label sequences in the frequency domain, FreDF mitigates the bias caused by label autocorrelation while maintaining the advantages of the DFT, including efficient inference and multi-task learning capabilities. Additionally, FreDF is model-agnostic, compatible with various forecasting models $g$ (e.g., Transformers and MLPs). This flexibility significantly expands the potential applications of FreDF across diverse time series forecasting scenarios, where different forecasting models may demonstrate superior performance.

## 4 Experiments

To demonstrate the efficacy of FreDF, there are six aspects empirically investigated:

1. **Performance:** *Does FreDF work?* Section 4.2 compares FreDF with state-of-the-art baselines using public datasets. The long-term forecasting task is investigated in Section 4.2 and the short-term forecasting and imputation tasks are explored in Appendix E.1.

Table 1: Long-term forecasting performance.

| Models | FreDF (Ours) | | iTransformer (2024) | | FreTS (2023) | | TimesNet (2023) | | MICN (2023) | | TiDE (2023) | | DLinear (2023) | | FEDformer (2022) | | Autoformer (2021) | | Transformer (2017) | | TCN (2017) | |
|---|---|---|---|---|---|---|---|---|---|---|---|---|---|---|---|---|---|---|---|---|---|---|
| Metrics | MSE | MAE | MSE | MAE | MSE | MAE | MSE | MAE | MSE | MAE | MSE | MAE | MSE | MAE | MSE | MAE | MSE | MAE | MSE | MAE | MSE | MAE |
| ETTm1 | 0.392 | 0.399 | 0.415 | 0.416 | 0.407 | 0.415 | 0.413 | 0.418 | 0.399 | 0.423 | 0.419 | 0.419 | 0.404 | 0.407 | 0.440 | 0.451 | 0.596 | 0.517 | 0.943 | 0.733 | 0.891 | 0.632 |
| ETTm2 | 0.278 | 0.319 | 0.294 | 0.335 | 0.335 | 0.379 | 0.297 | 0.332 | 0.300 | 0.356 | 0.358 | 0.404 | 0.344 | 0.396 | 0.302 | 0.348 | 0.326 | 0.366 | 1.322 | 0.814 | 3.411 | 1.432 |
| ETTh1 | 0.437 | 0.435 | 0.449 | 0.447 | 0.488 | 0.474 | 0.478 | 0.466 | 0.525 | 0.515 | 0.628 | 0.574 | 0.462 | 0.458 | 0.441 | 0.457 | 0.476 | 0.477 | 0.993 | 0.788 | 0.763 | 0.636 |
| ETTh2 | 0.371 | 0.396 | 0.390 | 0.410 | 0.550 | 0.515 | 0.413 | 0.426 | 0.624 | 0.549 | 0.611 | 0.550 | 0.558 | 0.516 | 0.430 | 0.447 | 0.478 | 0.483 | 3.296 | 1.419 | 3.325 | 1.445 |
| ECL | 0.170 | 0.259 | 0.176 | 0.267 | 0.209 | 0.297 | 0.214 | 0.307 | 0.187 | 0.297 | 0.251 | 0.344 | 0.225 | 0.319 | 0.229 | 0.339 | 0.228 | 0.339 | 0.274 | 0.367 | 0.617 | 0.598 |
| Traffic | 0.421 | 0.279 | 0.428 | 0.286 | 0.552 | 0.348 | 0.535 | 0.309 | 0.636 | 0.335 | 0.760 | 0.473 | 0.673 | 0.419 | 0.611 | 0.379 | 0.637 | 0.399 | 0.680 | 0.376 | 1.001 | 0.652 |
| Weather | 0.254 | 0.274 | 0.281 | 0.302 | 0.255 | 0.299 | 0.262 | 0.288 | 0.261 | 0.319 | 0.271 | 0.320 | 0.265 | 0.317 | 0.311 | 0.361 | 0.349 | 0.391 | 0.632 | 0.552 | 0.584 | 0.572 |
| PEMS03 | 0.113 | 0.219 | 0.116 | 0.226 | 0.146 | 0.257 | 0.118 | 0.223 | 0.099 | 0.214 | 0.316 | 0.370 | 0.233 | 0.344 | 0.174 | 0.302 | 0.501 | 0.513 | 0.126 | 0.233 | 0.666 | 0.634 |
| PEMS08 | 0.141 | 0.238 | 0.159 | 0.258 | 0.174 | 0.277 | 0.154 | 0.245 | 0.717 | 0.459 | 0.319 | 0.378 | 0.294 | 0.377 | 0.232 | 0.322 | 0.630 | 0.572 | 0.249 | 0.266 | 0.713 | 0.629 |

*Note*: We fix the input length as 96 following the established benchmark (Liu et al., 2024). **Bold** typeface highlights the top performance for each metric, while underlined text denotes the second-best results. The results are averaged over forecast lengths (96, 192, 336 and 720), with full results in Table 5.

2. **Mechanism:** *How does it work?* Section 4.3 offers an ablative study to dissect the contributions of FreDF's individual components, elucidating their roles in enhancing forecasting accuracy.

3. **Generality:** *Does it support other forecasting models?* Section 4.4 verifies the adaptability of FreDF across different forecasting models, with additional results documented in Appendix E.2.

4. **Flexibility:** *Does it support alternative transformations to FFT?* Section 4.4 replaces FFT with other transformations to showcase its flexibility of implementation.

5. **Sensitivity:** *Does it require careful fine-tuning?* Section 4.5 presents a sensitivity analysis of the hyperparameter $\alpha$, where FreDF maintains efficacy across a broad range of parameter values.

6. **Efficiency:** *Is FreDF effective given limited samples?* Section 4.6 offers a learning curve analysis, where FreDF achieves comparable performance with limited samples to that obtained using substantially more time-domain labels, indicating an advantageous sample efficiency.

## 4.1 SETUP

**Datasets.** The datasets for long-term forecast and imputation include ETT (4 subsets), ECL, Traffic, Weather and PEMS (Liu et al., 2024). The dataset for short-term forecast is M4 following Wu et al. (2023). Each dataset is divided chronologically for training, validation and test. Detailed dataset descriptions are provided in Appendix D.1.

**Baselines.** Our baselines include various established models, which can be grouped into three categories: (1) Transformer-based methods: Transformer (Vaswani et al., 2017), Autoformer (Wu et al., 2021), FEDformer (Zhou et al., 2022), iTransformer (Liu et al., 2024); (2) MLP-based methods: DLinear (Zeng et al., 2023), TiDE (Das et al., 2023), FreTS (Yi et al., 2023b); (3) other notable models: TimesNet (Wu et al., 2023), MICN (Wang et al., 2023b), TCN (Bai et al., 2018).

**Implementation.** The baseline models are reproduced using the scripts provided by Liu et al. (2024). They are trained using the Adam (Kingma & Ba, 2015) optimizer to minimize the MSE loss. Datasets are split chronologically into training, validation, and test sets. Following the protocol outlined in the comprehensive benchmark (Qiu et al., 2024), the dropping-last trick is disabled during the test phase. When integrating FreDF to enhance an established model, we adhere to the associated hyperparameter settings in the public benchmark (Liu et al., 2024), only tuning $\alpha$ and learning rate conservatively. Experiments are conducted on Intel(R) Xeon(R) Platinum 8383C CPUs and NVIDIA RTX 3090 GPUs. More implementation details are provided in Appendix D.2.

## 4.2 OVERALL PERFORMANCE

The performance on the long-term forecast task is presented in Table 1, where we select iTransformer as the forecast model $g$ and enhance it with FreDF. Overall, FreDF improves the performance of iTransformer substantially. For instance, on the ETTm1 dataset, FreDF decreases the MSE

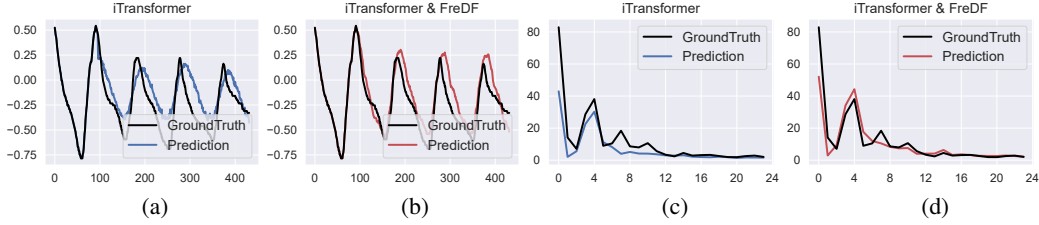

Figure 3: Visualization of forecast sequence generated with and without FreDF in the time (a-b) and frequency (c-d) domains, using the iTransformer as the backbone model.

Table 2: Ablation study results.

| Model | $\mathcal{L}^{(\text{tmp})}$ | $\mathcal{L}^{(\text{feq})}$ | Data | T=96 | | T=192 | | T=336 | | T=720 | | Avg | |
|---|---|---|---|---|---|---|---|---|---|---|---|---|---|
| | | | | MSE | MAE | MSE | MAE | MSE | MAE | MSE | MAE | MSE | MAE |
| DF | ✓ | ✗ | ETTm1 | 0.346 | 0.379 | 0.391 | 0.400 | 0.426 | 0.422 | 0.493 | 0.460 | 0.414 | 0.415 |
| | | | ETTh1 | 0.390 | 0.409 | 0.442 | 0.440 | 0.479 | 0.457 | 0.483 | 0.479 | 0.449 | 0.446 |
| | | | ECL | 0.147 | 0.239 | 0.166 | 0.258 | 0.178 | 0.271 | 0.209 | 0.298 | 0.175 | 0.266 |
| | | | Weather | 0.201 | 0.246 | 0.250 | 0.282 | 0.302 | 0.317 | 0.370 | 0.361 | 0.280 | 0.302 |
| FreDF† | ✗ | ✓ | ETTm1 | 0.324 | 0.361 | 0.374 | 0.387 | 0.403 | 0.405 | 0.468 | 0.443 | 0.392 | 0.399 |
| | | | ETTh1 | 0.380 | 0.399 | 0.429 | 0.425 | 0.474 | 0.451 | 0.467 | 0.464 | 0.437 | 0.435 |
| | | | ECL | 0.144 | 0.232 | 0.158 | 0.247 | 0.171 | 0.262 | 0.204 | 0.291 | 0.169 | 0.258 |
| | | | Weather | 0.165 | 0.205 | 0.225 | 0.255 | 0.278 | 0.295 | 0.359 | 0.349 | 0.257 | 0.276 |
| FreDF | ✓ | ✓ | ETTm1 | **0.324** | **0.362** | **0.372** | **0.385** | **0.402** | **0.404** | **0.468** | **0.443** | **0.391** | **0.398** |
| | | | ETTh1 | **0.381** | **0.400** | **0.430** | **0.426** | **0.474** | **0.451** | **0.463** | **0.461** | **0.437** | **0.435** |
| | | | ECL | **0.144** | **0.233** | **0.158** | **0.247** | **0.172** | **0.263** | **0.204** | **0.293** | **0.169** | **0.259** |
| | | | Weather | **0.163** | **0.202** | **0.220** | **0.252** | **0.274** | **0.293** | **0.356** | **0.346** | **0.253** | **0.273** |

of iTransformer by 0.019. Similar gains are evident in other datasets, which can be attributed to reconciliation of label autocorrelation with the DF paradigm, validating efficacy of FreDF.

Moreover, FreDF enhances the performance of iTransformer to surpass even those models that originally outperformed iTransformer on some datasets. It indicates that the improvements by FreDF exceed those achievable through dedicated architectural design alone, emphasizing the importance of handling label autocorrelation and FreDF.

**Showcases.** We visualize the forecast sequences to highlight the improvements of FreDF in forecast quality. An ETTm2 snapshot with T=336 is depicted in Fig. 3. Although the model without FreDF can follow the general trends of the label sequence, it struggles to capture the sequence's high-frequency components, resulting in a forecast with a visibly lower frequency. Additionally, the forecast sequence exhibits numerous burrs. These issues reflect the limitations of forecasting in the time domain, namely the difficulty in capturing high-frequency components and the neglect of autocorrelation between sequential steps. FreDF addresses these limitations effectively. The forecasts generated under FreDF not only keep pace with the label sequence, accurately capturing high-frequency components, but also exhibit a smoother appearance with fewer irregularities, due to its awareness of autocorrelation.

Table 3: Varying FFT implementation results.

| Model | ETTh1 | | | | ETTm1 | | | | ECL | | | |
|---|---|---|---|---|---|---|---|---|---|---|---|---|
| | MSE | Δ | MAE | Δ | MSE | Δ | MAE | Δ | MSE | Δ | MAE | Δ |
| iTransformer | 0.449 | - | 0.447 | - | 0.415 | - | 0.416 | - | 0.176 | - | 0.267 | - |
| + FreDF-T | 0.437 | ↓ 2.63% | 0.435 | ↓ 2.62% | 0.392 | ↓ 5.49% | 0.399 | ↓ 4.01% | 0.170 | ↓ 3.41% | 0.259 | ↓ 2.77% |
| + FreDF-D | 0.445 | ↓ 0.92% | 0.440 | ↓ 1.42% | 0.395 | ↓ 4.77% | 0.398 | ↓ 4.33% | 0.171 | ↓ 2.51% | 0.260 | ↓ 2.52% |
| + FreDF-2 | 0.432 | ↓ 3.94% | 0.431 | ↓ 3.57% | 0.392 | ↓ 5.60% | 0.399 | ↓ 4.05% | 0.166 | ↓ 5.32% | 0.256 | ↓ 4.20% |

*Note*: Δ denotes the relative error reduction compared to iTransformer with DF paradigm.

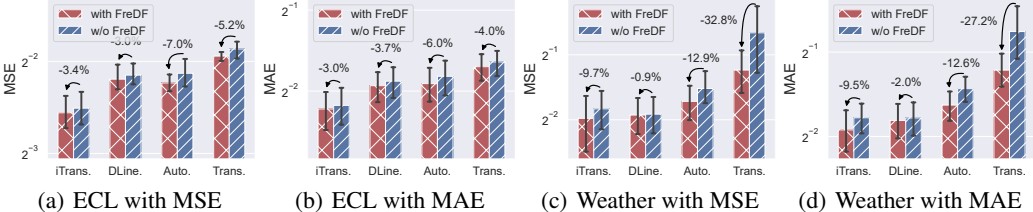

| (a) ECL with MSE | (b) ECL with MAE | (c) Weather with MSE | (d) Weather with MAE |

Figure 4: Benefit of incorporating FreDF in varying models, shown with colored bars for means over forecast lengths (96, 192, 336, 720) and error bars for 99.9% confidence intervals.

## 4.3 ABLATION STUDIES

In this section, we dissect the contributions of the temporal and frequency loss for enhancing forecast performance. The results are detailed in Table 2, where iTransformer is used as the forecast model. Overall, the frequency loss consistently improves performance compared to the temporal loss. The rationale is that label autocorrelation can be effectively managed in the frequency domain, aligning better with the conditional independence assumption inherent in DF. Moreover, learning to forecast in both domains generally showcase improvement compared to relying solely on one domain. However, the improvement over $\mathcal{L}^{(\mathrm{feq})}$ is marginal. Hence, exclusively focusing on frequency domain forecasting emerges as a viable strategy in most cases, offering promising performance without the complexity of balancing learning objectives.

## 4.4 GENERALIZATION STUDIES

In this section, we investigate the utility of FreDF with different forecast models and domain transformation strategies, to showcase the generality of FreDF. In the bar-plots, the forecast errors are averaged over forecast lengths (96, 192, 336, 720), with error bars as 95% confidence intervals.

**Varying forecast models.** We explore the versatility of FreDF in augmenting representative neural forecasting models: iTransformer, DLinear, Autoformer, and Transformer. FreDF demonstrates significant enhancements across these models compared to the traditional DF paradigm, as illustrated in Fig. 4. Notably, Transformer-based models such as the Autoformer and Transformer substantially benefit from the integration of FreDF. On the ECL dataset, for instance, the Autoformer (developed in 2021) enhanced by FreDF outperforms DLinear (developed in 2023). More evidence of FreDF's versatility is provided in Appendix E. These results confirm FreDF's potential as a plugin-and-play strategy to enhance various time series forecasting models.

**Varying FFT implementations.** We note that label autocorrelation exists between not only different steps, but also variables in multivariate forecasting. Therefore, we implement FFT along the time (FreDF-T) and variable dimension (FreDF-D) to handle the corresponding correlations, with the outcomes illustrated in Table 3. In general, conducting FFT along the time and variable axis brings similar performance gain, which showcases the existence of correlation between different steps and variables, respectively. In particular, FreDF-T slightly outperforms FreDF-D, which underscores the relative importance of auto-correlation in the label sequence. Finally, a strategic approach is viewing the multivariate sequence as an image, performing 2-dimensional FFT on both time and variable axes (FreDF-2), which accommodates the correlations between both time steps and variables simultaneously and further improves performance.

**Varying transformations.** Motivated by the fact that FFT can be viewed as projections onto exponential bases, we extend the implementation of FreDF by replacing FFT with projections onto other established polynomials. Each polynomial set is adept at capturing specific data patterns, such as trends and periodicity, which are challenging to learn in the time domain. The results are summarized in Fig. 5. Notably, projections onto Legendre and Fourier bases demonstrate superior performance. This superiority is attributed to the orthogonality of the polynomials, a feature not guaranteed by

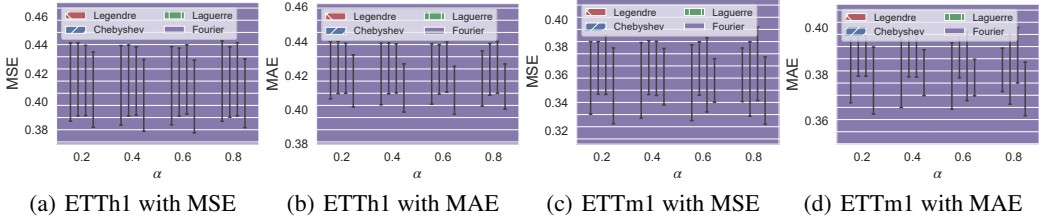

Figure 5: Varying projection bases results, shown with colored bars for means over forecast lengths (96, 192, 336, 720) and error bars for 99.9% confidence intervals.

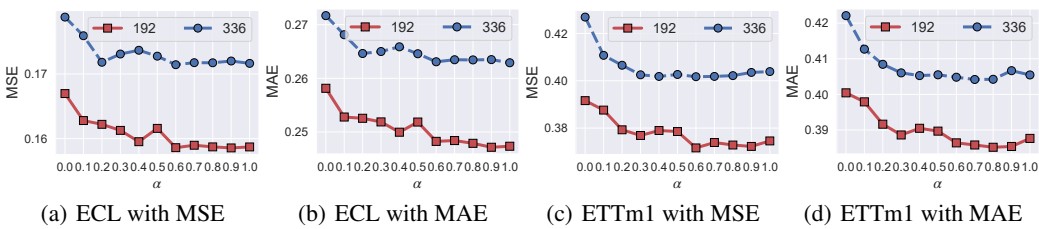

Figure 6: Varying strength of frequency loss ($\alpha$) results, shown with colored lines for T=192, 336.

others as analyzed in Appendix C. It underscores orthogonality when selecting polynomials for implementing FreDF, which is pivotal for eliminating autocorrelation.

## 4.5 HYPERPARAMETER SENSITIVITY

The key hyperparameter of FreDF is the frequency loss strength $\alpha$. The performance given different $\alpha$ is summarized in Fig. 6. Overall, increasing $\alpha$ from 0 to 1 results in a reduction of forecast error, albeit with a slight increase towards the end of this range. For instance, on the ECL dataset with T=192, both MAE and MSE decrease from approximately 0.258 and 0.167 to 0.247 and 0.158, respectively. Such trend of diminishing error seems consistent across different forecast lengths and datasets, supporting the benefit of learning to forecast in the frequency domain. Notably, the optimal reduction in forecast error typically occurs at $\alpha$ values near 1, such as 0.8 for the ETTh1 dataset, rather than at the absolute value of 1. Therefore, unifying supervision signals from both time and frequency domains brings performance improvement. Similar trends are presented across different datasets and foreacst models, as discussed in Appendix E.3.

## 4.6 LEARNING-CURVE ANALYSIS

In this section, we investigate the sample efficiency of learning in the time versus frequency domains, with the corresponding learning curves in Fig. 7. Overall, given limited training data, learning in the frequency domain demonstrates remarkable efficacy. With only 30% of the training data, it achieves performance comparable to learning in the time domain using the full training dataset.

The underlying reason for this enhanced sample efficiency can be attributed to the consistent and more straightforward nature of the data representation.

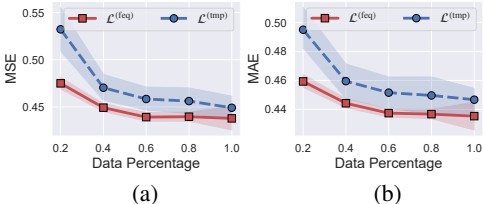

Figure 7: Learning curve on ETTm1 dataset.

For instance, a sliding window on a sine signal yields a set of distinct sequences in the time domain. However, in the frequency domain, these sequences present a similar pattern: a prominent spike at a specific frequency and negligible values elsewhere. This uniformity simplifies the learning process by

making patterns more consistent and easier to decipher, thus reducing the need for extensive training datasets.

## 5 CONCLUSION

In this study, we underscore the challenge of label autocorrelation in time series modeling, which biases the learning objective of the widely adopted DF paradigm. To tackle this challenge, we introduce a model-agnostic learning objective: FreDF, which mitigates label autocorrelation by transforming the label sequence into the frequency domain, thereby effectively reducing the bias caused by label autocorrelation. The experiments demonstrate that FreDF effectively enhances the performance of prevalent forecast models.

***Limitation & future works.*** In this work, we primarily utilize the Fourier transform for domain transformation. Despite empirical efficacy, the predefined set of exponential bases lacks the ability to adapt to specific data properties. Alternative transforms such independent component analysis can produce orthogonal bases considering data properties, representing a valuable avenue for future research. Additionally, the issue of label autocorrelation extends beyond time series, affecting diverse contexts involving structural labels, such as 3D point clouds, speech, and images. The potential of FreDF to enhance performance in these contexts awaits further exploration.

## ACKNOWLEDGEMENT

This work was supported by National Natural Science Foundation of China (623B2002, 12075212). The first author extends heartfelt gratitude to Prof. Degui Yang of Central South University, for his exceptional signal processing lectures and generous research guidance during S.T.E.M. studies.

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

## A    OVERVIEW OF DML FOR PARTIAL CORRELATION ESTIMATION

### A.1    MOTIVATION

In this section, we introduce the rationale for employing double machine learning (DML) to quantify the partial correlations. Our focus is on the autocorrelation represented by $Y_t \rightarrow Y_{t'}$ where $0 \leq t < t' < \mathrm{T}$. However, the fork structure $Y_t \leftarrow L(n) \rightarrow Y_{t'}$ creates a pseudo correlation between $Y_{t'}$ and $Y_t$ (Wang et al., 2024a). In this case, the autocorrelation $Y_t \rightarrow Y_{t'}$ is influenced by the pseudo correlations from the fork structure, rendering traditional correlation measures, such as Pearson correlation, ineffective for quantifying the autocorrelation $Y_t \rightarrow Y_{t'}$ (Li et al., 2024a;b).

To effectively address this influence and quantify partial correlation, it is essential to employ methods that excel in distinguishing direct relationships from spurious ones (Wang et al., 2023a). DML is chosen for calculating partial correlation due to its ease of implementation and independence from exhaustive hyperparameter tuning. DML offers a robust and reliable quantification of the autocorrelation that we care about (Bia et al., 2024; Chernozhukov et al., 2018).

### A.2    METHOD

In this section, we detail the implementation of DML, a two-step procedure designed for estimating partial correlation. We define $\mathcal{T} \in \mathbb{R}$ as the treatment variable, $\mathcal{Y} \in \mathbb{R}$ as the outcome variable, $\mathcal{X} \in \mathbb{R}^{\mathrm{D}}$ as the control variable that needs to be accounted for. The implementation of DML is depicted in Fig. 8 (b) which consists of two steps below.

- **Orthogonalization.** This step involves orthogonalizing both the outcome ($\mathcal{Y}$) and the treatment ($\mathcal{T}$) with respect to the control variables ($\mathcal{X}$). To this end, we first use two machine learning models, namely $\phi$ and $\psi$, to predict the outcome and the treatment based on $\mathcal{X}$. These predictions aim to capture the components in $\mathcal{Y}$ and $\mathcal{T}$ that are influenced by $\mathcal{X}$. Subsequently, such impact of $\mathcal{X}$ can be eliminated by calculating the residuals:

$$
\begin{aligned}
\tilde{\mathcal{Y}} &= \mathcal{Y} - \phi(\mathcal{X}), \\
\tilde{\mathcal{T}} &= \mathcal{T} - \psi(\mathcal{X}).
\end{aligned}
\tag{5}
$$

- **Regression.** This step involves regressing the orthogonalized outcome $\tilde{\mathcal{Y}}$ on the orthogonalized treatment $\tilde{\mathcal{T}}$. A linear regression model is utilized for this purpose:

$$
\tilde{\mathcal{Y}} = \beta \tilde{\mathcal{T}} + \epsilon,
\tag{6}
$$

where $\epsilon$ is the error term; $\beta$ is the model coefficient that can be identified via ordinary least squares. The $\beta$ can be identified in a supervised learning manner, with the objective of minimizing the MSE between the prediction and real values. The identified $\beta$ quantifies the partial correlation between the treatment and the outcome, having accounted for the influence of $\mathcal{X}$.

By regressing the orthogonalized outcome on the orthogonalized treatment, DML captures the direct effect of the treatment on the outcome without the interference from control variables, as depicted in Fig. 8 (c). That is, DML isolates the desired partial correlation $\mathcal{T} \rightarrow \mathcal{Y}$ from the influencing correlation $\mathcal{T} \leftarrow \mathcal{X} \rightarrow \mathcal{Y}$.

### A.3    EXPERIMENTAL SETTINGS

In this section, we outline the experimental settings implemented to employ DML for quantifying the correlations of interest.

**General settings.**    For the base learners $\phi$ and $\psi$, we opt for a linear regression model optimized using ordinary least squares for its efficiency[2]. Following Appendix A.1, we treat the input sequence $L$ as the control variable to adjust, and simplify the process by considering the last step in $L$ as representative. Moreover, we focus exclusively on the correlations within the last feature of each

---

[2]The linear regression model, chosen for its computational efficiency, is crucial in managing the experiment's scale, where the total number of DML estimators can be exceedingly high (e.g., 36,864 for T=192).

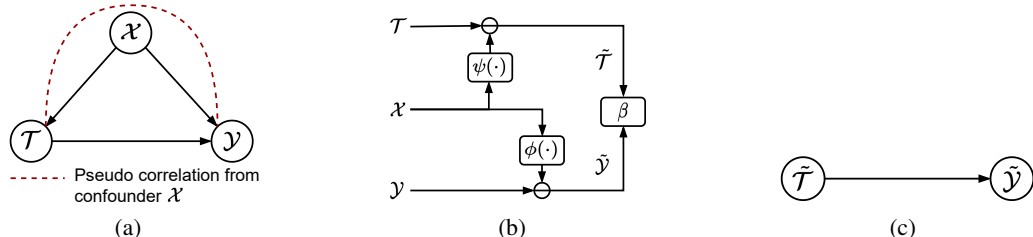

Figure 8: Visualization of partial correlation and DML approach for partial correlation quantification. (a) The correlation graph where the pseudo correlation is caused by the fork structure $\mathcal{T} \leftarrow \mathcal{X} \rightarrow \mathcal{Y}$. (b) The implementation of DML, where $\beta$ is the identified strength of the partial correlation $\mathcal{T} \rightarrow \mathcal{Y}$. (c) The partial correlation identified by DML.

dataset[3]. This focus makes $Y$ a scalar value within the real number space rather than a D-dimensional vector in this experiment.

**Specifications for identifying time-domain partial correlation.** To assess the partial correlation $Y_t \rightarrow Y_{t'}$, we treat $Y_t$ as the treatment and $Y_{t'}$ as the outcome. The DML model is trained using a set of N observations: $\{L(n)\}_{n=1:N}$, $\{Y_t(n)\}_{n=1:N}$, and $\{Y_{t'}(n)\}_{n=1:N}$. The coefficient $\beta$ derived from the DML model is interpreted as the strength of the partial correlation $Y_t \rightarrow Y_{t'}$.

**Specifications for identifying frequency-domain partial correlation.** To quantify the partial correlation $F_k \rightarrow F_{k'}$, we treat $F_k$ as the treatment and $F_{k'}$ as the outcome. The DML model is trained using a set of N observations: $\{L(n)\}_{n=1:N}$, $\{F_k(n)\}_{n=1:N}$, and $\{F_{k'}(n)\}_{n=1:N}$. The coefficient $\beta$ derived from the DML model is interpreted as the strength of the partial correlation $F_k \rightarrow F_{k'}$. A notable complexity arises because $F_k$ is a complex number. Since DML is typically designed for real numbers instead of complex numbers, it requires a separate consideration of the real and imaginary parts of $F_k$.

## A.4 MORE EXPERIMENTAL RESULTS

In this section, we provide comprehensive results of the identified partial correlation strengths, which quantifies the autocorrelation effect in the time and frequency domain. Fig. 9 presents the results on three different datasets: Traffic, ETTh1, and ECL, with forecast length set to 192. Fig. 10 presents the results for varying forecast lengths: 48, 96, 192, 336, on the ECL dataset.

The results show similar patterns to those in the main text. Specifically, the non-diagonal elements in Fig. 9 (a-c) and Fig. 10 (a-d) often exhibit huge values, which affirms the presence of label autocorrelation in the time domain. In contrast, the non-diagonal elements in Fig. 9 (d-i) and Fig. 10 (e-l) show negligible values, which suggests that frequency components of $F$ are almost independent given $L$. These findings collectively verify (1) the existence of label autocorrelation in the time domain; (2) the mitigation of label correlation in the frequency domain.

---

[3]This focus is aligned with the study's objective of analyzing autocorrelation instead of inter-feature correlations, which simplifies the interpretation of results.

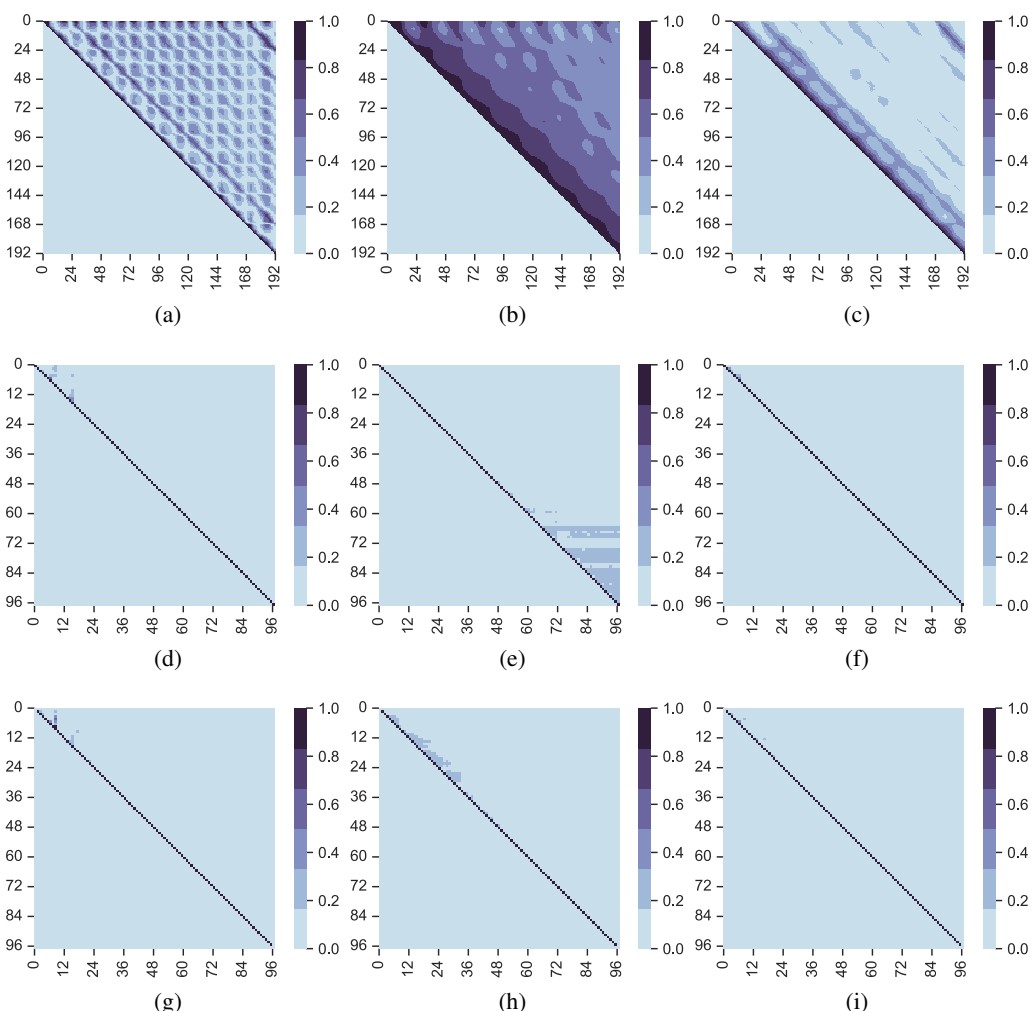

Figure 9: More comprehensive visualizations of label autocorrelation in different domains and datasets, with columns representing different datasets: Traffic, ETTh1, and ECL, from left to right. Panels (a-c) show the label correlation in the time domain, where each element $\rho_{i,j}$ indicates the partial correlation between $Y_i$ and $Y_j$ given $L$. Panels (d-i) show the label correlation in the frequency domain, where each element $\rho_{i,j}$ indicates the partial correlation between $F_i$ and $F_j$ given $L$, shown with the real (d-f) and imaginary part (g-i). Due to the symmetry inherent in FFT, the forecast length in the frequency domain is halved.

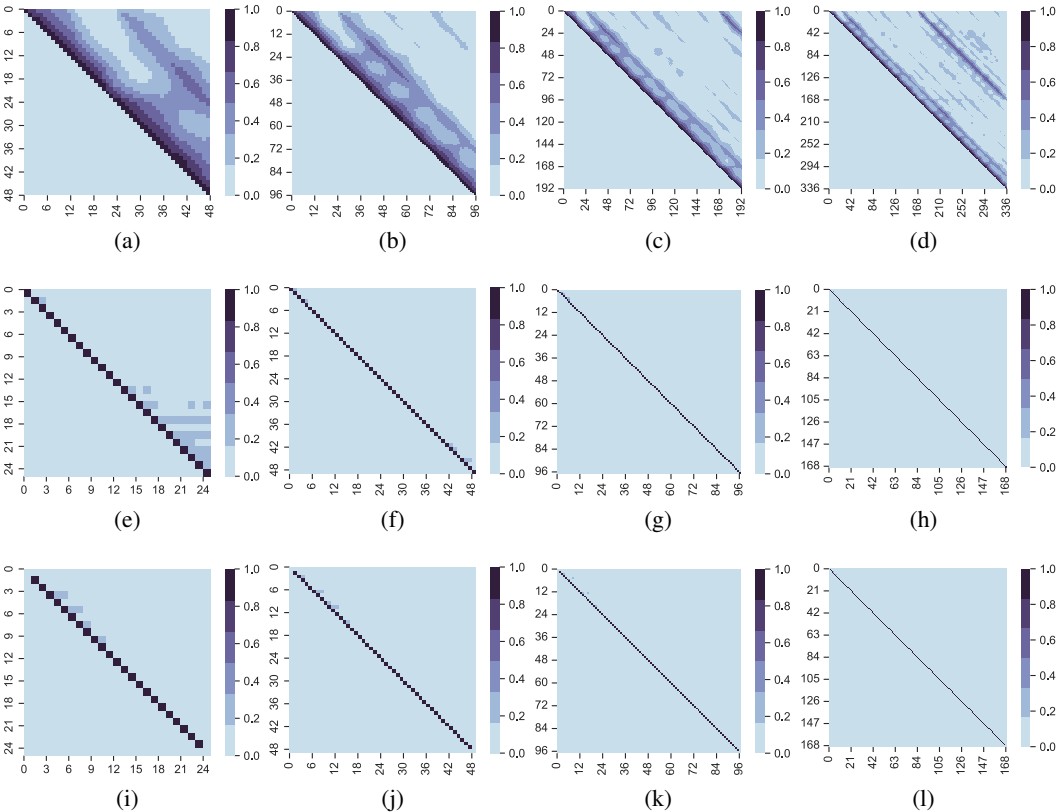

Figure 10: More comprehensive visualizations of label autocorrelation in different domains and label lengths, with columns representing label lengths H=48, 96, 192, 336 from left to right. Panels (a-d) show the label correlation in the time domain, where each element $\rho_{i,j}$ indicates the partial correlation between $Y_i$ and $Y_j$ given $L$. Panels (e-l) show the label correlation in the frequency domain, where each element $\rho_{i,j}$ indicates the partial correlation between $F_i$ and $F_j$ given $L$, shown with the real (e-h) and imaginary part (i-l).

## B    THEORETICAL JUSTIFICATION

**Theorem B.1** (Bias of vanilla DF, simplified). *Given an input sequence $L$ and a univariate label sequence $Y = [Y_1, Y_2]$ (the forecast length is set to 2 for simplicity), the learning objective* (1) *of the DF paradigm is biased against the practical NLL, expressed as:*

$$\text{Bias} = \frac{1}{2\sigma^2}(Y_2 - \hat{Y}_2)^2 - \frac{1}{2\sigma^2(1-\rho^2)}(Y_2 - (\hat{Y}_2 + \rho(Y_1 - \hat{Y}_1)))^2, \tag{7}$$

*where $\hat{Y}_i$ indicates the prediction at the $i$-th step and $\rho$ denotes the partial correlation between $Y_1$ and $Y_2$ given $L$.*

*Proof.* Aligning with the maximum likelihood analysis, we assume the label sequence obeys a normal distribution with mean $\mu = [\hat{Y}_1, \hat{Y}_2]$ and covariance $\zeta = [[\sigma^2, \rho\sigma^2], [\rho\sigma^2, \sigma_2^2]]$. The negative log-likelihood (NLL) of $Y$ given the input sequence $L$ can be expressed as

$$-\log p(Y|L) = -\log p(Y_1|L) - \log p(Y_2|L, Y_1)$$

$$= -\log(\frac{1}{\sqrt{2\pi}\sigma}\exp(-\frac{(Y_1 - \hat{Y}_1)^2}{2\sigma^2}))$$

$$-\log(\frac{1}{\sqrt{2\pi(1-\rho^2)}\sigma}\exp(-\frac{(Y_2 - (\hat{Y}_2 + \rho(Y_1 - \hat{Y}_1)))^2}{2\sigma^2(1-\rho^2)})).$$

Removing coefficients unrelated to $g$, the practical NLL that contributes the gradients to update $g$ is

$$\text{NLL} := \frac{1}{2\sigma^2}(Y_1 - \hat{Y}_1)^2 + \frac{1}{2\sigma^2(1-\rho^2)}(Y_2 - (\hat{Y}_2 + \rho(Y_1 - \hat{Y}_1)))^2.$$

If the independence assumption of different time step holds (i.e., $Y_1$ and $Y_2$ are conditionally independent given $L$), we have $\rho = 0$, followed by $p(Y_2|L, Y_1) = p(Y_2|L)$. In this case, the MSE loss in canonical DF mirrors the practical NLL:

$$\text{MSE} = \frac{1}{2\sigma^2}(Y_1 - \hat{Y}_1)^2 + \frac{1}{2\sigma^2}(Y_2 - \hat{Y}_2)^2,$$

where $\sigma$ is often set to 1 when implementing MSE. If the independence assumption does not hold, i.e., considering autocorrelation in the label sequence, we have $\rho \neq 0$. In this case, the MSE loss in the time domain is biased to the practical NLL, expressed as:

$$\text{Bias} = \frac{1}{2\sigma^2}(Y_2 - \hat{Y}_2)^2 - \frac{1}{2\sigma^2(1-\rho^2)}(Y_2 - (\hat{Y}_2 + \rho(Y_1 - \hat{Y}_1)))^2.$$

This bias introduced by label autocorrelation makes the MSE loss in the time domain fail to reflect the practical NLL and therefore misleads the update of forecast model $g$ under DF paradigm.    □

**Theorem B.2** (Bias of vanillia DF). *Given an input sequence $L$ and a univariate label sequence $Y$, the learning objective* (1) *of the DF paradigm is biased against the practical NLL, expressed as:*

$$\text{Bias} = \sum_{i=1}^{\text{T}}\frac{1}{2\sigma^2}(Y_i - \hat{Y}_i)^2 - \sum_{i=1}^{\text{T}}\frac{1}{2\sigma^2(1-\rho_i^2)}\left(Y_i - \left(\hat{Y}_i + \sum_{j=1}^{i-1}\rho_{ij}(Y_j - \hat{Y}_j)\right)\right)^2, \tag{8}$$

*where $\hat{Y}_i$ indicates the prediction at the $i$-th step, $\rho_{ij}$ denotes the partial correlation between $Y_i$ and $Y_j$ given $L$, $\rho_i^2 = \sum_{j=1}^{i-1}\rho_{ij}^2$.*

*Proof.* We assume that the label sequence $Y$ conditioned on the input sequence $L$ follows a multivariate normal distribution with mean vector $\mu = [\hat{Y}_1, \hat{Y}_2, \ldots, \hat{Y}_{\text{T}}]$ and covariance matrix $\Sigma$, where the diagonal entries $\Sigma_{ii} = \sigma^2$ and the off-diagonal entries are $\Sigma_{ij} = \rho_{ij}\sigma^2$ for $i \neq j$. Here, $\rho_{ij}$ denotes the partial correlation between $Y_i$ and $Y_j$ given the input sequence $L$. On the basis, the NLL of the

label sequence $Y$ given $L$ can be decomposed into a sum of conditional NLLs due to the properties of the multivariate normal distribution:

$$-\log p(Y \mid L) = -\sum_{i=1}^{T} \log p(Y_i \mid L, Y_1, Y_2, \ldots, Y_{i-1}),$$

where each conditional probability $p(Y_i \mid L, Y_1, \ldots, Y_{i-1})$ is Gaussian with mean $\hat{Y}_i + \sum_{j=1}^{i-1} \rho_{ij}(Y_j - \hat{Y}_j)$ and variance $\sigma^2(1 - \rho_i^2)$, $\rho_i^2 = \sum_{j=1}^{i-1} \rho_{ij}^2$. Thus, the NLL can be expressed as

$$-\log p(Y \mid L) = \sum_{i=1}^{T} \left( \frac{1}{2} \log(2\pi\sigma^2(1 - \rho_i^2)) + \frac{1}{2\sigma^2(1 - \rho_i^2)} \left( Y_i - \left( \hat{Y}_i + \sum_{j=1}^{i-1} \rho_{ij}(Y_j - \hat{Y}_j) \right) \right)^2 \right).$$

For the purpose of gradient-based optimization, terms independent of the model predictions $\hat{Y}_i$ can be omitted. Therefore, the practical NLL contributing to the gradients is given by

$$\text{NLL} = \sum_{i=1}^{T} \frac{1}{2\sigma^2(1 - \rho_i^2)} \left( Y_i - \left( \hat{Y}_i + \sum_{j=1}^{i-1} \rho_{ij}(Y_j - \hat{Y}_j) \right) \right)^2.$$

On the other hand, the DF paradigm typically employs the MSE loss, expressed as

$$\text{MSE} = \sum_{i=1}^{T} \frac{1}{2\sigma^2}(Y_i - \hat{Y}_i)^2.$$

which deviates from the practical NLL. The bias is expressed as:

$$\text{Bias} = \text{MSE} - \text{NLL} = \sum_{i=1}^{T} \frac{1}{2\sigma^2}(Y_i - \hat{Y}_i)^2 - \sum_{i=1}^{T} \frac{1}{2\sigma^2(1 - \rho_i^2)} \left( Y_i - \hat{Y}_i + \sum_{j=1}^{i-1} \rho_{ij}(Y_j - \hat{Y}_j) \right)^2.$$

When there exists label autocorrelation, i.e., $\rho_{ij} \neq 0$, the bias above exists. *In the special case where the label autocorrelation is diminished, i.e., $\rho_{ij} \to 0$, the bias approaches zero almost surely.*

$\square$

**Corollary B.3** (Bias of vanilla DF, multivariate). *Given an input sequence $L$ and a multivariate label sequence $Y \in \mathbb{R}^{T \times D}$, suppose $Z \in \mathbb{R}^{T \times D}$ is the flattened version of $Y$ obtained by concatenating the rows, the learning objective (1) of the DF paradigm is biased against the practical NLL:*

$$\text{Bias} = \sum_{i=1}^{T \times D} \frac{1}{2\sigma^2}(Z_i - \hat{Z}_i)^2 - \sum_{i=1}^{T \times D} \frac{1}{2\sigma^2(1 - \rho_i^2)} \left( Z_i - \left( \hat{Z}_i + \sum_{j=1}^{i-1} \rho_{ij}(Z_j - \hat{Z}_j) \right) \right)^2, \quad (9)$$

*where $\hat{Z}_i$ indicates the prediction of $Z_i$, $\rho_{ij}$ denotes the partial correlation between $Z_i$ and $Z_j$ given $L$, $\rho_i^2 = \sum_{j=1}^{i-1} \rho_{ij}^2$.*

*Proof.* This corollary immediately follows from Theorem B.2, by viewing the multivariate label sequence $Z$ as an augmented univariate sequence. $\square$

**Theorem B.4** (Decorrelation between frequency components). *Suppose $Y$ is a zero-mean, discrete-time, wide-sense stationary random process of length $T$. As $T \to \infty$, the normalized DFT coefficients become asymptotically uncorrelated at different frequencies:*

$$\lim_{T \to \infty} \mathbb{E}[F_k F_{k'}^*] = \begin{cases} S_Y(f_k), & \text{if } k = k', \\ 0, & \text{if } k \neq k', \end{cases}$$

*where $f_k = \frac{k}{T}$ and $S_Y(f)$ is the power spectral density of $Y$.*

*Proof.* Recalling that the normalized DFT coefficients $F_k$ are defined as $F_k = 1/\sqrt{\mathrm{T}} \sum_{t=0}^{\mathrm{T}-1} Y_t e^{-j2\pi kt/\mathrm{T}}$, $k = 0, 1, \ldots, \mathrm{T}-1$. On this basis, the expected value of the product $F_k F_{k'}^*$ can be expressed as:

$$
\begin{aligned}
\mathbb{E}[F_k F_{k'}^*] &= \mathbb{E}\left[\sum_{t=0}^{\mathrm{T}-1} Y_t e^{-j2\pi kt/\mathrm{T}} \cdot \sum_{t'=0}^{\mathrm{T}-1} Y_{t'} e^{j2\pi k't'/\mathrm{T}}\right]/\mathrm{T} \\
&= \sum_{t=0}^{\mathrm{T}-1}\sum_{t'=0}^{\mathrm{T}-1} R_Y[t-t'] e^{-j2\pi kt/\mathrm{T}} e^{j2\pi k't'/\mathrm{T}}/\mathrm{T},
\end{aligned}
\tag{10}
$$

where we interchanged the order of summation and expectation, and utilize the autocorrelation function $R_Y[\tau] = \mathbb{E}[Y_t Y_{t'}]$. Denote $\tau = t - t'$, which allows us to rewrite $t' = t - \tau$. This substitution leads us to:

$$
\begin{aligned}
\mathbb{E}[F_k F_{k'}^*] &= \sum_{t=0}^{\mathrm{T}-1}\sum_{\tau=t}^{t-\mathrm{T}+1} R_Y[\tau] e^{-j2\pi(kt/\mathrm{T}-k'(t-\tau)/\mathrm{T})}/\mathrm{T} \\
&= \sum_{\tau=-(\mathrm{T}-1)}^{\mathrm{T}-1} R_Y[\tau] e^{-j2\pi k'\tau/\mathrm{T}} \left(\sum_{t=\max(0,\tau)}^{\min(\mathrm{T}-1,\mathrm{T}-1+\tau)} e^{j2\pi(k'-k)t/\mathrm{T}}/\mathrm{T}\right).
\end{aligned}
$$

which immediately follows switching the order of summation. The expression within the parentheses is a summation of complex exponentials. When $k \neq k'$, the inner term approaches zero due to the mutual cancellation of the oscillatory exponentials:

$$
\lim_{\mathrm{T}\to\infty} \mathbb{E}[F_k F_{k'}^*] = 0.
$$

When $k = k'$, the exponential term becomes unity, and the inner sum simplifies to:

$$
\lim_{\mathrm{T}\to\infty} \sum_{t=\max(0,\tau)}^{\min(\mathrm{T}-1,\mathrm{T}-1+\tau)} 1/\mathrm{T} = \lim_{\mathrm{T}\to\infty} 1 - |\tau|/\mathrm{T} = 1.
$$

which immediately follows by $\mathbb{E}[F_k F_{k'}^*] = S_Y(f_k)$, where $S_Y$ is the power spectral density of $Y$ that can be calculated as the DFT of $R_Y$. The proof is therefore completed. $\square$

## C  GENERALIZED TRANSFORMATION ONTO DIFFERENT BASES

Transforming time series data onto predefined spaces is a fundamental aspect of signal processing and data analysis, with various strategies available based on the selected bases, such as Fourier and Chebyshev bases. The selection of bases is determined by the specific characteristics and requirements of the analysis. Below, we present formal definitions of common transform techniques and their associated bases, where we formulate signals as continuous functions for the ease of demonstration.

**Fourier transform.**  It employs exponential polynominals as bases which prove to be mutually orthogonal. These polynomials are effective for analyzing periodic signals or signals with a strong frequency component. Let $k$ be the frequency, the associated basis function and projection onto it can be formulated as follows:

$$
\begin{aligned}
f_k(t) &= \exp(-j(2\pi/\mathrm{H})kt), \\
F_k &= \int_{-\infty}^{\infty} x(t) f_k(t) dt
\end{aligned}
\tag{11}
$$

**Legendre transform.**  It uses the Legendre polynomials as bases which prove to be mutually orthogonal on the interval $[-1, 1]$. These polynomials are particularly useful for representing functions defined on a finite interval, which makes them suitable for certain types of data smoothing and approximation tasks. The $k$-th polynomial and the associated projection can be formulated as:

$$
\begin{aligned}
f_k(t) &= \frac{1}{2^k k!} \frac{d^k}{dt^k}[(t^2-1)^k], \\
F_k &= \int_{-1}^{1} x(t) f_k(t) dt
\end{aligned}
\tag{12}
$$

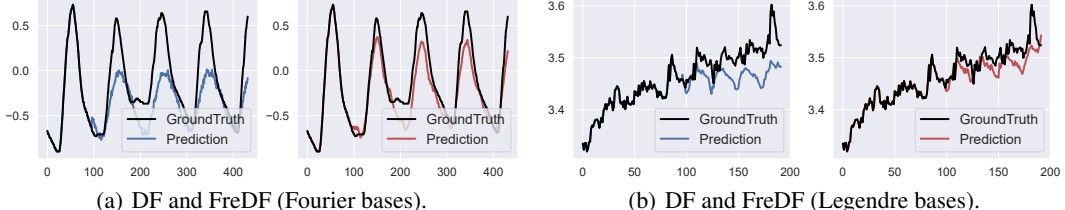

(a) DF and FreDF (Fourier bases).  (b) DF and FreDF (Legendre bases).

Figure 11: The label sequences (black lines) and forecast sequences generated by DF (blue lines) and FreDF (red lines). The forecast model used is iTransformer, with experiments conducted on selected snapshots characterized by periodicity (a) and trend (b).

**Chebyshev transform.** It uses the Chebyshev polynomials as bases. These bases are not originally orthogonal but become mutually orthogonal on the interval $[-1, 1]$ with respect to the weight $1/\sqrt{1 - t^2}$. These polynomials are particularly useful for approximating functions with rapid variations. The $k$-th Chebyshev polynomial and the associated projection can be formulated as follows:

$$f_k(t) = \cos(k \arccos(t)),$$
$$F_k = \int_{-1}^{1} \frac{x(t)f_k(t)}{\sqrt{1 - t^2}} dt \tag{13}$$

**Laguerre transform.** It uses the Laguerre polynomials as bases. These bases are NOT originally orthogonal but become mutually orthogonal on the interval $[0, \infty]$ with respect to the exponential weight $\exp(t)$. These polynomials are particularly useful in quantum mechanics and other fields involving exponential decay. The $k$-th Laguerre polynomial and the associated projection can be formulated as follows:

$$f_k(t) = \exp(t)\frac{d^k}{dt^k}(\exp(-t)t^k),$$
$$F_k = \int_{0}^{\infty} \frac{x(t)f_k(t)}{\exp(t)} dt \tag{14}$$

These polynomial sets are effective for capturing specific data patterns, such as trends and periodicity, which can be difficult to learn in the time domain. By incorporating these polynomial sets, FreDF enhances its flexibility to handle time series data with varying characteristics. A case study is presented in Fig. 11. Specifically, the forecast sequences generated by the canonical DF struggle to capture increasing trends or high-frequency periods; whereas those produced by FreDF effectively capture the dominant characteristics, thereby significantly improving forecast quality.

*In summary, FreDF does not rely solely on Fourier bases but can be adapted to various bases, each with unique properties suitable for different applications. The selection of bases for FreDF depends on the characteristics of the data and the specific objectives of the analysis.*

## D  REPRODUCTION DETAILS

### D.1  DATASET DESCRIPTIONS

The datasets utilized in this study cover a wide range of time series data, detailed in Table 4, each exhibiting unique characteristics and temporal resolutions:

- ETT (Li et al., 2021) comprises data on 7 factors related to electricity transformers, collected from July 2016 to July 2018. This dataset is divided into four subsets: ETTh1 and ETTh2, with hourly recordings, and ETTm1 and ETTm2, documented every 15 minutes.
- Weather (Wu et al., 2021) includes 21 meteorological variables gathered every 10 minutes throughout 2020 from the Weather Station of the Max Planck Biogeochemistry Institute.
- ECL (Electricity Consumption Load) (Wu et al., 2021) presents hourly electricity consumption data for 321 clients.

Table 4: Dataset description.

| Dataset | D | Forecast length | Train / validation / test | Frequency | Domain |
|---|---|---|---|---|---|
| ETTh1 | 7 | 96, 192, 336, 720 | 8545/2881/2881 | Hourly | Health |
| ETTh2 | 7 | 96, 192, 336, 720 | 8545/2881/2881 | Hourly | Health |
| ETTm1 | 7 | 96, 192, 336, 720 | 34465/11521/11521 | 15min | Health |
| ETTm2 | 7 | 96, 192, 336, 720 | 34465/11521/11521 | 15min | Health |
| Weather | 21 | 96, 192, 336, 720 | 36792/5271/10540 | 10min | Weather |
| ECL | 321 | 96, 192, 336, 720 | 18317/2633/5261 | Hourly | Electricity |
| Traffic | 862 | 96, 192, 336, 720 | 12185/1757/3509 | Hourly | Transportation |
| PEMS03 | 358 | 12, 24, 36, 48 | 15617/5135/5135 | 5min | Transportation |
| PEMS08 | 170 | 12, 24, 36, 48 | 10690/3548/265 | 5min | Transportation |

*Note*: *D* denotes the number of variates. *Frequency* denotes the sampling interval of time points. *Train, Validation, Test* denotes the number of samples employed in each split. The taxonomy aligns with Wu et al. (2023).

- Traffic (Wu et al., 2021) features hourly road occupancy rates from 862 sensors in the San Francisco Bay area freeways, spanning from January 2015 to December 2016.
- PEMS (Liu et al., 2022a) contains the public traffic network data in California collected by 5-minute windows. Two public subsets (PEMS03, PEMS08) are adopted in this work.

The datasets are chronologically divided into training, validation, and test sets following the protocols outlined in (Qiu et al., 2024; Liu et al., 2024). The dropping-last trick is disabled during the test phase. The length of the input sequence is standardized at 96 across the ETT, Weather, ECL, and Traffic datasets, with varying label sequence lengths of 96, 192, 336, and 720.

### D.2 IMPLEMENTATION DETAILS

The baseline models in this study are reproduced using training scripts obtained from the iTransformer repository (Liu et al., 2024) after reproducibility verification. Models are trained using the Adam optimizer (Kingma & Ba, 2015), with learning rates selected from the set $10^{-3}, 5 \times 10^{-4}, 10^{-4}$ to minimize the MSE loss. The training is limited to a maximum of 10 epochs, incorporating an early stopping mechanism activated upon a lack of improvement in validation performance over 3 epochs.

In experiments integrating FreDF to enhance an existing forecast model, we adhere to the associated hyperparameter settings from the public benchmark (Liu et al., 2024), tuning only $\alpha$ within $[0, 1]$ and learning rate conservatively. Finetuning the learning rate is essential to handle the different magnitude of temporal and frequency losses. Fine-tuning is conducted to minimize the MSE averaged across all forecast lengths on the validation dataset.

## E MORE EXPERIMENTAL RESULTS

### E.1 OVERALL PERFORMANCE

**Long-term forecast.** We provide comprehensive performance comparison on the long-term forecast task in Table 5. The iTransformer model is used to operationalize the FreDF paradigm. Despite the iTransformer's existing performance gap compared to other baseline models, the incorporation of FreDF enhances its performance in the majority of cases, securing the lowest MSE in 31 out of 45 cases and MAE in 40 out of 45 cases. The few instances where FreDF does not achieve the lowest MSE are attributed to the inherent superiority of other models over iTransformer in specific datasets (for example, FreTS versus iTransformer on the Weather dataset).

**Short-term forecast.** We investigate the short-term forecast task in Table 6, with FreTS Yi et al. (2023b) serving as the forecasting model in the FreDF implementation. Consistent with the long-term forecasting results, FreDF enhances FreTS's performance in most instances. Notably, there are three

Table 5: The comprehensive results on the long-term forecasting task.

| Models | | FreDF (Ours) | | iTransformer (2024) | | FreTS (2023) | | TimesNet (2023) | | MICN (2023) | | TiDE (2023) | | DLinear (2023) | | FEDformer (2022) | | Autoformer (2021) | | Transformer (2017) | | TCN (2017) | |
|---|---|---|---|---|---|---|---|---|---|---|---|---|---|---|---|---|---|---|---|---|---|---|---|
| Metrics | | MSE | MAE | MSE | MAE | MSE | MAE | MSE | MAE | MSE | MAE | MSE | MAE | MSE | MAE | MSE | MAE | MSE | MAE | MSE | MAE | MSE | MAE |
| ETTm1 | 96 | 0.324 | 0.362 | 0.346 | 0.379 | 0.339 | 0.374 | 0.338 | 0.379 | 0.318 | 0.366 | 0.364 | 0.387 | 0.345 | 0.372 | 0.389 | 0.427 | 0.468 | 0.463 | 0.591 | 0.549 | 0.887 | 0.613 |
| | 192 | 0.373 | 0.385 | 0.392 | 0.400 | 0.382 | 0.397 | 0.389 | 0.400 | 0.364 | 0.396 | 0.398 | 0.404 | 0.381 | 0.390 | 0.402 | 0.431 | 0.573 | 0.509 | 0.704 | 0.629 | 0.877 | 0.626 |
| | 336 | 0.402 | 0.404 | 0.427 | 0.422 | 0.421 | 0.426 | 0.429 | 0.428 | 0.398 | 0.428 | 0.428 | 0.425 | 0.414 | 0.414 | 0.438 | 0.451 | 0.596 | 0.527 | 1.171 | 0.861 | 0.890 | 0.636 |
| | 720 | 0.469 | 0.444 | 0.494 | 0.461 | 0.485 | 0.462 | 0.495 | 0.464 | 0.514 | 0.501 | 0.487 | 0.461 | 0.473 | 0.451 | 0.529 | 0.498 | 0.749 | 0.569 | 1.307 | 0.893 | 0.911 | 0.653 |
| | Avg | 0.392 | 0.399 | 0.415 | 0.416 | 0.407 | 0.415 | 0.413 | 0.418 | 0.399 | 0.423 | 0.419 | 0.419 | 0.404 | 0.407 | 0.440 | 0.451 | 0.596 | 0.517 | 0.943 | 0.733 | 0.891 | 0.632 |
| ETTm2 | 96 | 0.173 | 0.252 | 0.184 | 0.266 | 0.190 | 0.282 | 0.185 | 0.264 | 0.178 | 0.275 | 0.207 | 0.305 | 0.195 | 0.294 | 0.194 | 0.284 | 0.240 | 0.319 | 0.317 | 0.408 | 3.125 | 1.345 |
| | 192 | 0.241 | 0.298 | 0.257 | 0.315 | 0.260 | 0.329 | 0.254 | 0.307 | 0.240 | 0.317 | 0.290 | 0.364 | 0.283 | 0.359 | 0.264 | 0.324 | 0.300 | 0.349 | 1.069 | 0.758 | 3.130 | 1.350 |
| | 336 | 0.298 | 0.334 | 0.315 | 0.351 | 0.373 | 0.405 | 0.314 | 0.345 | 0.299 | 0.354 | 0.377 | 0.422 | 0.384 | 0.427 | 0.319 | 0.359 | 0.339 | 0.375 | 1.325 | 0.869 | 3.185 | 1.375 |
| | 720 | 0.398 | 0.393 | 0.419 | 0.409 | 0.517 | 0.499 | 0.434 | 0.413 | 0.482 | 0.479 | 0.558 | 0.502 | 0.516 | 0.502 | 0.420 | 0.424 | 0.423 | 0.421 | 2.576 | 1.223 | 4.203 | 1.658 |
| | Avg | 0.278 | 0.319 | 0.294 | 0.335 | 0.335 | 0.379 | 0.297 | 0.332 | 0.300 | 0.356 | 0.358 | 0.404 | 0.344 | 0.396 | 0.302 | 0.348 | 0.326 | 0.366 | 1.322 | 0.814 | 3.411 | 1.432 |
| ETTh1 | 96 | 0.382 | 0.400 | 0.390 | 0.410 | 0.399 | 0.412 | 0.422 | 0.433 | 0.383 | 0.418 | 0.479 | 0.464 | 0.396 | 0.410 | 0.377 | 0.418 | 0.423 | 0.441 | 0.796 | 0.691 | 0.767 | 0.633 |
| | 192 | 0.430 | 0.427 | 0.443 | 0.441 | 0.453 | 0.443 | 0.465 | 0.457 | 0.500 | 0.491 | 0.521 | 0.503 | 0.449 | 0.444 | 0.421 | 0.445 | 0.498 | 0.485 | 0.813 | 0.699 | 0.739 | 0.619 |
| | 336 | 0.474 | 0.451 | 0.480 | 0.457 | 0.503 | 0.475 | 0.492 | 0.470 | 0.546 | 0.530 | 0.659 | 0.603 | 0.487 | 0.465 | 0.468 | 0.472 | 0.506 | 0.496 | 1.181 | 0.876 | 0.717 | 0.613 |
| | 720 | 0.463 | 0.462 | 0.484 | 0.479 | 0.596 | 0.565 | 0.532 | 0.502 | 0.671 | 0.620 | 0.893 | 0.736 | 0.516 | 0.513 | 0.500 | 0.493 | 0.477 | 0.487 | 1.182 | 0.885 | 0.828 | 0.678 |
| | Avg | 0.437 | 0.435 | 0.449 | 0.447 | 0.488 | 0.474 | 0.478 | 0.466 | 0.525 | 0.515 | 0.628 | 0.574 | 0.462 | 0.458 | 0.441 | 0.457 | 0.476 | 0.477 | 0.993 | 0.788 | 0.763 | 0.636 |
| ETTh2 | 96 | 0.289 | 0.337 | 0.301 | 0.349 | 0.350 | 0.403 | 0.320 | 0.364 | 0.361 | 0.404 | 0.400 | 0.440 | 0.343 | 0.396 | 0.347 | 0.391 | 0.383 | 0.424 | 2.072 | 1.140 | 3.171 | 1.364 |
| | 192 | 0.363 | 0.385 | 0.382 | 0.402 | 0.472 | 0.475 | 0.409 | 0.417 | 0.495 | 0.490 | 0.528 | 0.509 | 0.473 | 0.474 | 0.430 | 0.443 | 0.557 | 0.511 | 5.081 | 1.814 | 3.222 | 1.398 |
| | 336 | 0.419 | 0.426 | 0.430 | 0.434 | 0.564 | 0.528 | 0.449 | 0.451 | 0.671 | 0.588 | 0.643 | 0.571 | 0.603 | 0.546 | 0.469 | 0.475 | 0.470 | 0.481 | 3.564 | 1.475 | 3.306 | 1.452 |
| | 720 | 0.415 | 0.437 | 0.447 | 0.455 | 0.815 | 0.654 | 0.473 | 0.474 | 0.968 | 0.712 | 0.874 | 0.679 | 0.812 | 0.650 | 0.473 | 0.480 | 0.501 | 0.515 | 2.469 | 1.247 | 3.599 | 1.565 |
| | Avg | 0.371 | 0.396 | 0.390 | 0.410 | 0.550 | 0.515 | 0.413 | 0.426 | 0.624 | 0.549 | 0.611 | 0.550 | 0.558 | 0.516 | 0.430 | 0.447 | 0.478 | 0.483 | 3.296 | 1.419 | 3.325 | 1.445 |
| ECL | 96 | 0.144 | 0.233 | 0.148 | 0.239 | 0.189 | 0.277 | 0.171 | 0.273 | 0.168 | 0.280 | 0.237 | 0.329 | 0.210 | 0.302 | 0.200 | 0.315 | 0.199 | 0.315 | 0.252 | 0.352 | 0.688 | 0.621 |
| | 192 | 0.159 | 0.247 | 0.167 | 0.258 | 0.193 | 0.282 | 0.188 | 0.289 | 0.177 | 0.289 | 0.236 | 0.330 | 0.210 | 0.305 | 0.207 | 0.322 | 0.215 | 0.327 | 0.266 | 0.366 | 0.587 | 0.582 |
| | 336 | 0.172 | 0.263 | 0.179 | 0.272 | 0.207 | 0.296 | 0.208 | 0.304 | 0.185 | 0.296 | 0.249 | 0.344 | 0.223 | 0.319 | 0.226 | 0.340 | 0.232 | 0.343 | 0.292 | 0.383 | 0.590 | 0.588 |
| | 720 | 0.204 | 0.294 | 0.209 | 0.298 | 0.245 | 0.332 | 0.289 | 0.363 | 0.218 | 0.323 | 0.284 | 0.373 | 0.258 | 0.350 | 0.282 | 0.379 | 0.268 | 0.371 | 0.287 | 0.371 | 0.602 | 0.601 |
| | Avg | 0.170 | 0.259 | 0.176 | 0.267 | 0.209 | 0.297 | 0.214 | 0.307 | 0.187 | 0.297 | 0.251 | 0.344 | 0.225 | 0.319 | 0.229 | 0.339 | 0.228 | 0.339 | 0.274 | 0.367 | 0.617 | 0.598 |
| Traffic | 96 | 0.391 | 0.265 | 0.397 | 0.272 | 0.528 | 0.341 | 0.504 | 0.298 | 0.609 | 0.317 | 0.805 | 0.493 | 0.697 | 0.429 | 0.577 | 0.362 | 0.609 | 0.385 | 0.686 | 0.385 | 1.451 | 0.744 |
| | 192 | 0.410 | 0.273 | 0.418 | 0.279 | 0.531 | 0.338 | 0.526 | 0.305 | 0.621 | 0.328 | 0.756 | 0.474 | 0.647 | 0.407 | 0.603 | 0.372 | 0.633 | 0.400 | 0.679 | 0.377 | 0.842 | 0.622 |
| | 336 | 0.424 | 0.280 | 0.432 | 0.286 | 0.551 | 0.345 | 0.540 | 0.310 | 0.641 | 0.342 | 0.762 | 0.477 | 0.653 | 0.410 | 0.615 | 0.378 | 0.637 | 0.398 | 0.663 | 0.361 | 0.844 | 0.620 |
| | 720 | 0.460 | 0.298 | 0.467 | 0.305 | 0.598 | 0.367 | 0.570 | 0.324 | 0.671 | 0.354 | 0.719 | 0.449 | 0.694 | 0.429 | 0.649 | 0.403 | 0.668 | 0.415 | 0.693 | 0.381 | 0.867 | 0.624 |
| | Avg | 0.421 | 0.279 | 0.428 | 0.286 | 0.552 | 0.348 | 0.535 | 0.309 | 0.636 | 0.335 | 0.760 | 0.473 | 0.673 | 0.419 | 0.611 | 0.379 | 0.637 | 0.399 | 0.680 | 0.376 | 1.001 | 0.652 |
| Weather | 96 | 0.164 | 0.202 | 0.201 | 0.247 | 0.184 | 0.239 | 0.178 | 0.226 | 0.182 | 0.250 | 0.202 | 0.261 | 0.197 | 0.259 | 0.221 | 0.304 | 0.284 | 0.355 | 0.332 | 0.383 | 0.610 | 0.568 |
| | 192 | 0.220 | 0.253 | 0.250 | 0.283 | 0.223 | 0.275 | 0.227 | 0.266 | 0.234 | 0.301 | 0.242 | 0.298 | 0.236 | 0.294 | 0.275 | 0.345 | 0.313 | 0.371 | 0.634 | 0.539 | 0.541 | 0.552 |
| | 336 | 0.275 | 0.294 | 0.302 | 0.317 | 0.272 | 0.316 | 0.283 | 0.305 | 0.268 | 0.325 | 0.287 | 0.335 | 0.282 | 0.332 | 0.338 | 0.379 | 0.359 | 0.393 | 0.656 | 0.579 | 0.565 | 0.569 |
| | 720 | 0.356 | 0.347 | 0.370 | 0.362 | 0.340 | 0.363 | 0.359 | 0.355 | 0.361 | 0.399 | 0.351 | 0.386 | 0.347 | 0.384 | 0.408 | 0.418 | 0.440 | 0.446 | 0.908 | 0.706 | 0.622 | 0.601 |
| | Avg | 0.254 | 0.274 | 0.281 | 0.302 | 0.255 | 0.299 | 0.262 | 0.288 | 0.261 | 0.319 | 0.271 | 0.320 | 0.265 | 0.317 | 0.311 | 0.361 | 0.349 | 0.391 | 0.632 | 0.552 | 0.584 | 0.572 |
| PEMS03 | 12 | 0.068 | 0.172 | 0.069 | 0.175 | 0.083 | 0.194 | 0.082 | 0.188 | 0.087 | 0.203 | 0.117 | 0.225 | 0.122 | 0.245 | 0.123 | 0.248 | 0.239 | 0.365 | 0.107 | 0.209 | 0.632 | 0.606 |
| | 24 | 0.096 | 0.205 | 0.098 | 0.210 | 0.127 | 0.241 | 0.110 | 0.216 | 0.086 | 0.198 | 0.233 | 0.320 | 0.202 | 0.320 | 0.160 | 0.287 | 0.492 | 0.506 | 0.121 | 0.227 | 0.655 | 0.626 |
| | 36 | 0.128 | 0.240 | 0.131 | 0.243 | 0.169 | 0.281 | 0.133 | 0.236 | 0.105 | 0.220 | 0.380 | 0.422 | 0.275 | 0.382 | 0.191 | 0.321 | 0.399 | 0.459 | 0.133 | 0.243 | 0.678 | 0.644 |
| | 48 | 0.161 | 0.269 | 0.164 | 0.275 | 0.204 | 0.311 | 0.146 | 0.251 | 0.120 | 0.235 | 0.536 | 0.511 | 0.335 | 0.429 | 0.223 | 0.350 | 0.875 | 0.723 | 0.144 | 0.253 | 0.699 | 0.659 |
| | Avg | 0.113 | 0.219 | 0.116 | 0.226 | 0.146 | 0.257 | 0.118 | 0.223 | 0.099 | 0.214 | 0.316 | 0.370 | 0.233 | 0.344 | 0.174 | 0.302 | 0.501 | 0.513 | 0.126 | 0.233 | 0.666 | 0.634 |
| PEMS08 | 12 | 0.080 | 0.182 | 0.085 | 0.189 | 0.095 | 0.204 | 0.110 | 0.209 | 2.193 | 0.871 | 0.121 | 0.231 | 0.152 | 0.274 | 0.175 | 0.275 | 0.446 | 0.483 | 0.213 | 0.236 | 0.680 | 0.607 |
| | 24 | 0.118 | 0.220 | 0.131 | 0.236 | 0.150 | 0.259 | 0.142 | 0.239 | 0.235 | 0.339 | 0.232 | 0.326 | 0.245 | 0.350 | 0.211 | 0.305 | 0.488 | 0.509 | 0.238 | 0.256 | 0.701 | 0.622 |
| | 36 | 0.161 | 0.258 | 0.182 | 0.282 | 0.202 | 0.305 | 0.167 | 0.258 | 0.197 | 0.300 | 0.379 | 0.428 | 0.344 | 0.417 | 0.250 | 0.338 | 0.532 | 0.513 | 0.263 | 0.277 | 0.727 | 0.637 |
| | 48 | 0.206 | 0.293 | 0.236 | 0.323 | 0.250 | 0.341 | 0.195 | 0.274 | 0.242 | 0.324 | 0.543 | 0.527 | 0.437 | 0.469 | 0.293 | 0.371 | 1.052 | 0.781 | 0.283 | 0.295 | 0.746 | 0.648 |
| | Avg | 0.141 | 0.238 | 0.159 | 0.258 | 0.174 | 0.277 | 0.154 | 0.245 | 0.717 | 0.459 | 0.319 | 0.378 | 0.294 | 0.377 | 0.232 | 0.322 | 0.630 | 0.572 | 0.249 | 0.266 | 0.713 | 0.629 |
| 1st Count | | 31 | 40 | 0 | 0 | 1 | 0 | 1 | 1 | 10 | 4 | 0 | 0 | 0 | 0 | 3 | 0 | 0 | 0 | 0 | 0 | 0 | 0 |

*Note*: We fix the input length as 96 following (Liu et al., 2024). **Bold** typeface highlights the top performance for each metric, while underlined text denotes the second-best results. *Avg* indicates the results averaged over forecasting lengths: T=96, 192, 336 and 720.

Table 6: The comprehensive results on the short-term forecasting task.

| Models | FreDF (Ours) | | | FreTS (2023) | | | iTransformer (2024) | | | MICN (2023) | | | DLinear (2023) | | | Fedformer (2023) | | | Autoformer (2023) | | |
|---|---|---|---|---|---|---|---|---|---|---|---|---|---|---|---|---|---|---|---|---|---|
| Metric | SMAPE | MASE | OWA | SMAPE | MASE | OWA | SMAPE | MASE | OWA | SMAPE | MASE | OWA | SMAPE | MASE | OWA | SMAPE | MASE | OWA | SMAPE | MASE | OWA |
| Yearly | 13.556 | 3.046 | 0.798 | 13.576 | 3.068 | 0.801 | 13.797 | 3.143 | 0.818 | 14.594 | 3.392 | 0.873 | 14.307 | 3.094 | 0.827 | 13.648 | 3.089 | 0.806 | 18.477 | 4.26 | 1.101 |
| Quarterly | 10.374 | 1.229 | 0.919 | 10.361 | 1.223 | 0.916 | 10.503 | 1.248 | 0.932 | 11.417 | 1.385 | 1.023 | 10.500 | 1.237 | 0.928 | 10.612 | 1.246 | 0.936 | 14.254 | 1.829 | 1.314 |
| Monthly | 12.999 | 0.983 | 0.913 | 13.088 | 0.99 | 0.919 | 13.227 | 1.013 | 0.935 | 13.834 | 1.080 | 0.987 | 13.362 | 1.007 | 0.937 | 14.181 | 1.105 | 1.011 | 18.421 | 1.616 | 1.398 |
| Others | 5.294 | 3.614 | 1.127 | 5.563 | 3.71 | 1.17 | 5.101 | 3.419 | 1.076 | 6.137 | 4.201 | 1.308 | 5.12 | 3.649 | 1.114 | 4.823 | 3.243 | 1.019 | 6.772 | 4.963 | 1.495 |
| Avg. | 12.112 | 1.648 | 0.877 | 12.169 | 1.66 | 0.883 | 12.298 | 1.68 | 0.893 | 13.044 | 1.841 | 0.962 | 12.48 | 1.674 | 0.898 | 12.734 | 1.702 | 0.914 | 16.851 | 2.443 | 1.26 |
| 1st Count | 3 | 3 | 3 | 1 | 1 | 1 | 0 | 0 | 0 | 0 | 0 | 0 | 0 | 0 | 0 | 1 | 1 | 1 | 0 | 0 | 0 |

*Note*: **Bold** typeface highlights the top performance for each metric, while underlined text denotes the second-best results. *Avg* indicates the results averaged over forecasting lengths: yearly, quarterly, and monthly.

Table 7: The comprehensive results on the missing data imputation task.

| Models | FreDF (Ours) | | iTransformer (2024) | | FreTS (2023) | | TimesNet (2023) | | MICN (2023) | | TiDE (2023) | | DLinear (2023) | | FEDformer (2022) | | Autoformer (2021) | |
|---|---|---|---|---|---|---|---|---|---|---|---|---|---|---|---|---|---|---|
| $p_{miss}$ | MSE | MAE | MSE | MAE | MSE | MAE | MSE | MAE | MSE | MAE | MSE | MAE | MSE | MAE | MSE | MAE | MSE | MAE |
| **ETTm1** 0.125 | 0.00153 | 0.02790 | 0.00213 | 0.03307 | 0.01102 | 0.07843 | 0.01152 | 0.07267 | 0.00236 | 0.03371 | 0.45052 | 0.45514 | 0.00148 | 0.02380 | 0.68262 | 0.38111 | 0.37654 | 0.35378 |
| 0.25 | 0.00287 | 0.03801 | 0.00402 | 0.04434 | 0.01089 | 0.07753 | 0.01245 | 0.07946 | 0.00284 | 0.03691 | 0.41777 | 0.45884 | 0.00154 | 0.02351 | 0.68235 | 0.38116 | 0.37059 | 0.35261 |
| 0.375 | 0.00256 | 0.03669 | 0.00458 | 0.04663 | 0.01100 | 0.07812 | 0.01407 | 0.08673 | 0.00323 | 0.03900 | 0.62935 | 0.55570 | 0.00175 | 0.02385 | 0.68191 | 0.38105 | 0.37877 | 0.36093 |
| 0.5 | 0.00152 | 0.02739 | 0.00363 | 0.04359 | 0.01102 | 0.07818 | 0.01676 | 0.09610 | 0.00352 | 0.04028 | 0.29342 | 0.39320 | 0.00192 | 0.02219 | 0.68119 | 0.38085 | 0.38052 | 0.36462 |
| Avg | 0.00212 | 0.03250 | 0.00359 | 0.04191 | 0.01098 | 0.07807 | 0.01370 | 0.08374 | 0.00299 | 0.03747 | 0.44776 | 0.46572 | 0.00167 | 0.02334 | 0.68202 | 0.38104 | 0.37660 | 0.35798 |
| **ETTm2** 0.125 | 0.00363 | 0.03840 | 0.00398 | 0.04034 | 0.03194 | 0.13349 | 0.01189 | 0.06710 | 0.00219 | 0.03345 | 0.83023 | 0.62174 | 0.03822 | 0.12943 | 3.10388 | 1.31356 | 1.40160 | 0.80777 |
| 0.25 | 0.00437 | 0.04255 | 0.00431 | 0.04303 | 0.03591 | 0.13655 | 0.01795 | 0.08939 | 0.00331 | 0.04100 | 0.81402 | 0.61100 | 0.03063 | 0.11547 | 3.10364 | 1.31348 | 1.41033 | 0.81363 |
| 0.375 | 0.00352 | 0.03823 | 0.00342 | 0.03793 | 0.03250 | 0.13336 | 0.02742 | 0.11499 | 0.00431 | 0.04598 | 1.11225 | 0.73633 | 0.01709 | 0.08822 | 3.10328 | 1.31330 | 1.40812 | 0.81049 |
| 0.5 | 0.00137 | 0.02382 | 0.00160 | 0.02538 | 0.03126 | 0.13027 | 0.04053 | 0.14285 | 0.00505 | 0.04918 | 0.99459 | 0.70665 | 0.01025 | 0.06440 | 3.10527 | 1.31389 | 1.44617 | 0.81796 |
| Avg | 0.00322 | 0.03575 | 0.00333 | 0.03667 | 0.03290 | 0.13342 | 0.02445 | 0.10358 | 0.00371 | 0.04240 | 0.93777 | 0.66893 | 0.02405 | 0.09938 | 3.10402 | 1.31356 | 1.41655 | 0.81246 |
| **ETTh1** 0.125 | 0.00178 | 0.03059 | 0.00319 | 0.04102 | 0.01400 | 0.08181 | 0.00441 | 0.04403 | 0.00432 | 0.04655 | 0.36363 | 0.45350 | 0.00279 | 0.03617 | 0.68307 | 0.38026 | 0.43136 | 0.41184 |
| 0.25 | 0.00218 | 0.03405 | 0.00334 | 0.04205 | 0.01347 | 0.08097 | 0.00320 | 0.03850 | 0.00454 | 0.04769 | 0.28435 | 0.40516 | 0.00236 | 0.03324 | 0.68162 | 0.37973 | 0.43515 | 0.41584 |
| 0.375 | 0.00182 | 0.03108 | 0.00280 | 0.03852 | 0.01308 | 0.08017 | 0.00261 | 0.03540 | 0.00454 | 0.04730 | 0.21038 | 0.34029 | 0.00210 | 0.03121 | 0.68181 | 0.37975 | 0.44431 | 0.42505 |
| 0.5 | 0.00114 | 0.02414 | 0.00174 | 0.03008 | 0.01276 | 0.07918 | 0.00245 | 0.03472 | 0.00437 | 0.04594 | 0.13344 | 0.27102 | 0.00175 | 0.02844 | 0.68137 | 0.37992 | 0.44312 | 0.42387 |
| Avg | 0.00173 | 0.02996 | 0.00277 | 0.03792 | 0.01333 | 0.08053 | 0.00317 | 0.03817 | 0.00444 | 0.04687 | 0.24795 | 0.36749 | 0.00225 | 0.03226 | 0.68197 | 0.37992 | 0.43848 | 0.41915 |
| **ETTh2** 0.125 | 0.00222 | 0.03124 | 0.00473 | 0.04606 | 0.04485 | 0.13849 | 0.00535 | 0.04495 | 0.00334 | 0.04202 | 1.15859 | 0.73871 | 0.02287 | 0.10885 | 3.12756 | 1.31746 | 1.45130 | 0.84467 |
| 0.25 | 0.00407 | 0.04258 | 0.00571 | 0.05096 | 0.04647 | 0.13551 | 0.00494 | 0.04476 | 0.00457 | 0.04950 | 0.75643 | 0.59747 | 0.02491 | 0.11511 | 3.12891 | 1.31754 | 1.45386 | 0.84388 |
| 0.375 | 0.00306 | 0.03693 | 0.00452 | 0.04519 | 0.04830 | 0.13583 | 0.00512 | 0.04697 | 0.00535 | 0.05363 | 0.59470 | 0.52371 | 0.01944 | 0.10277 | 3.12788 | 1.31728 | 1.45464 | 0.84194 |
| 0.5 | 0.00129 | 0.02365 | 0.00249 | 0.03304 | 0.04900 | 0.13469 | 0.00604 | 0.05224 | 0.00584 | 0.05547 | 0.35775 | 0.40497 | 0.01465 | 0.08746 | 3.12882 | 1.31733 | 1.45997 | 0.84644 |
| Avg | 0.00266 | 0.03360 | 0.00436 | 0.04381 | 0.04715 | 0.13613 | 0.00536 | 0.04723 | 0.00477 | 0.05016 | 0.71687 | 0.56622 | 0.02046 | 0.10355 | 3.12829 | 1.31740 | 1.45494 | 0.84423 |
| **ECL** 0.125 | 0.00029 | 0.01257 | 0.00187 | 0.03191 | 0.01018 | 0.08255 | 0.00466 | 0.04597 | 0.03678 | 0.14078 | 0.32942 | 0.42254 | 0.10658 | 0.23808 | 0.45884 | 0.41005 | 0.20147 | 0.29003 |
| 0.25 | 0.00061 | 0.01846 | 0.00216 | 0.03491 | 0.01022 | 0.08269 | 0.00341 | 0.03978 | 0.04106 | 0.14847 | 0.28831 | 0.40031 | 0.10682 | 0.23654 | 0.45887 | 0.41007 | 0.20618 | 0.29771 |
| 0.375 | 0.00090 | 0.02242 | 0.00211 | 0.03473 | 0.01022 | 0.08258 | 0.00230 | 0.03296 | 0.04373 | 0.15224 | 0.25310 | 0.37626 | 0.10500 | 0.23415 | 0.45886 | 0.41006 | 0.20998 | 0.30337 |
| 0.5 | 0.00103 | 0.02393 | 0.00175 | 0.03177 | 0.01025 | 0.08284 | 0.00171 | 0.02856 | 0.04520 | 0.15380 | 0.21280 | 0.34526 | 0.10362 | 0.23127 | 0.45891 | 0.41011 | 0.21322 | 0.30764 |
| Avg | 0.00071 | 0.01935 | 0.00197 | 0.03333 | 0.01022 | 0.08266 | 0.00302 | 0.03682 | 0.04169 | 0.14882 | 0.27091 | 0.38609 | 0.10550 | 0.23501 | 0.45887 | 0.41007 | 0.20771 | 0.29969 |
| **Weather** 0.125 | 0.00050 | 0.01259 | 0.00061 | 0.01446 | 0.00661 | 0.06123 | 0.00300 | 0.02110 | 0.00317 | 0.03646 | 0.36982 | 0.40486 | 0.00514 | 0.05275 | 0.40556 | 0.42631 | 0.13538 | 0.17599 |
| 0.25 | 0.00067 | 0.01513 | 0.00073 | 0.01715 | 0.00657 | 0.06105 | 0.00214 | 0.01830 | 0.00325 | 0.03900 | 0.29296 | 0.36483 | 0.00476 | 0.05019 | 0.40558 | 0.42635 | 0.13688 | 0.18177 |
| 0.375 | 0.00054 | 0.01443 | 0.00067 | 0.01700 | 0.00658 | 0.06113 | 0.00088 | 0.00924 | 0.00326 | 0.03997 | 0.17569 | 0.28913 | 0.00454 | 0.04811 | 0.40550 | 0.42633 | 0.13831 | 0.18700 |
| 0.5 | 0.00031 | 0.01107 | 0.00047 | 0.01429 | 0.00650 | 0.06071 | 0.00042 | 0.00463 | 0.00309 | 0.03929 | 0.12578 | 0.24598 | 0.00492 | 0.04961 | 0.40551 | 0.42632 | 0.13850 | 0.19051 |
| Avg | 0.00051 | 0.01331 | 0.00062 | 0.01573 | 0.00656 | 0.06103 | 0.00161 | 0.01332 | 0.00320 | 0.03868 | 0.24106 | 0.32620 | 0.00484 | 0.05016 | 0.40554 | 0.42633 | 0.13727 | 0.18382 |
| 1st Count | **23** | **19** | 1 | 1 | 0 | 0 | 0 | 2 | 2 | 2 | 0 | 0 | 4 | 6 | 0 | 0 | 0 | 0 |

*Note*: The input length is set to 96 for all baselines. **Bold** typeface highlights the top performance for each metric, while underlined text denotes the second-best results. *Avg* indicates the results averaged over missing ratios: 0.125, 0.25, 0.375, 0.5.

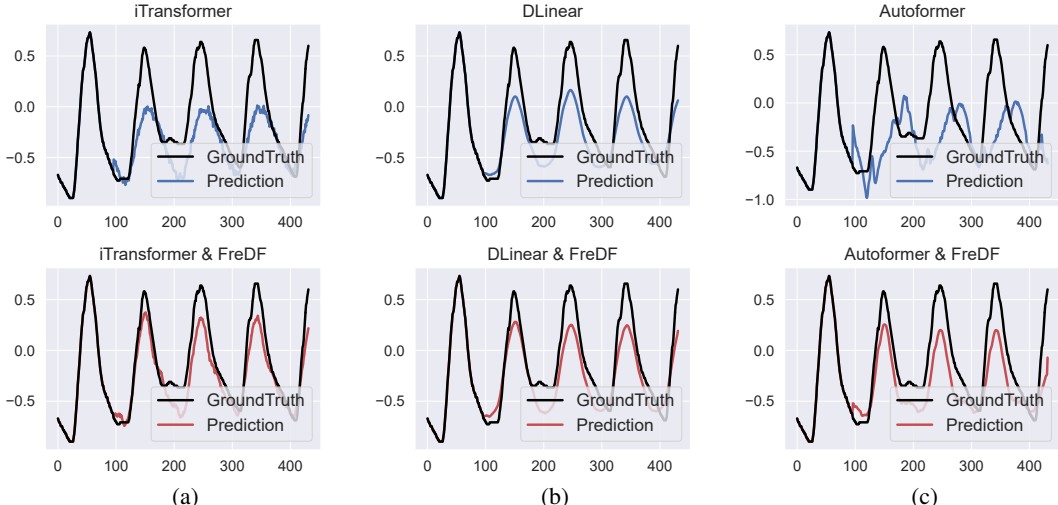

Figure 12: The forecast sequences generated with DF and FreDF. The forecast length is set to 336 and the experiment is conducted on a snapshot of ETTm2.

cases where FreTS outperforms FreDF. This occurs because the loss weight $\alpha$ is tuned to minimize the validation error averaged across all forecast lengths instead of focusing on specific lengths. While it is feasible to fine-tune $\alpha$ for each forecast length, we did not use this approach, as the current results suffice to demonstrate FreDF's effectiveness.

**Missing data imputation.** We investigate the imputation task in Table 7, with iTransformer serving as the forecasting model in the FreDF implementation. All models are trained using an autoencoding approach: given input sequences with missing entries, they are tasked with recovering the non-missing entries during training, while they are employed to impute the missing entries during inference. The results demonstrate FreDF's efficacy in this task, significantly improving the performance of iTransformer and outperforming most competitive methods. A unique aspect of this task is the irregularity of the label sequences caused by the missing entries, which disrupts the physical semantics related to the Fourier transform. This indicates that the effectiveness of FreDF does not stem from the semantic characteristics of the Fourier transform itself, but rather from its ability to align the properties of time series data with the implicit assumptions of the DF paradigm, specifically the conditional independence of labels.

**Showcases.** We provide additional showcases illustrating the change of forecast sequences in Fig. 12 and 14. Overall, FreDF effectively mitigates blurs and captures high frequency components. These successes can be attributed to FreDF's unique capability to operate in the frequency domain, where the challenges of autocorrelation are mitigated, and the expression of high-frequency components becomes straightforward.

### E.2 GENERALIZATION STUDIES

In this section, we further explore the versatility of FreDF in improving various forecasting models: iTransformer, DLinear, Autoformer, and Transformer. The results, displayed in Fig. 16, encompass five distinct datasets and are averaged over forecast lengths (96, 192, 336, 720), with error bars reflecting 95% confidence intervals. FreDF significantly improves the performance of these forecasting models, particularly benefiting Transformer-based architectures like Autoformer and Transformer. These results affirm FreDF's utility in enhancing neural forecasting models, highlighting its potential as a versatile training methodology in time series forecasting.

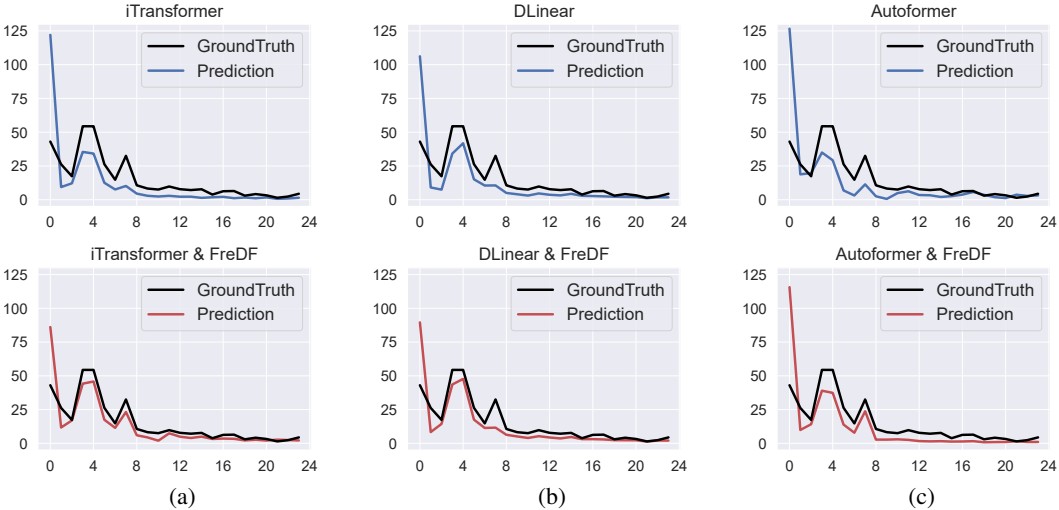

Figure 13: The spectrum of forecast sequences generated with DF and FreDF. The forecast length is set to 336 and the experiment is conducted on a snapshot of ETTm2. Only the first 24 frequencies of the spectrum are presented.

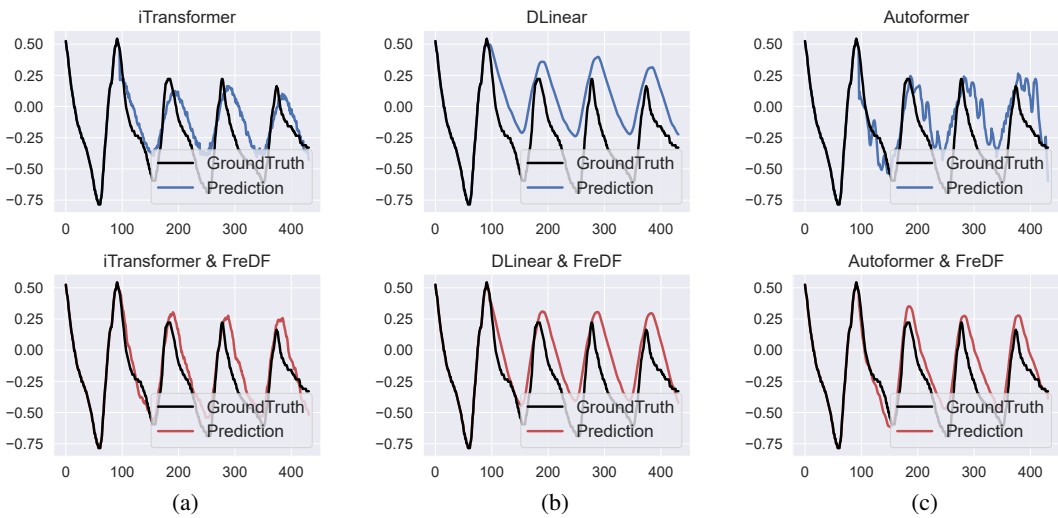

Figure 14: The forecast sequences generated with DF and FreDF. The forecast length is set to 336 and the experiment is conducted on another snapshot of ETTm2.

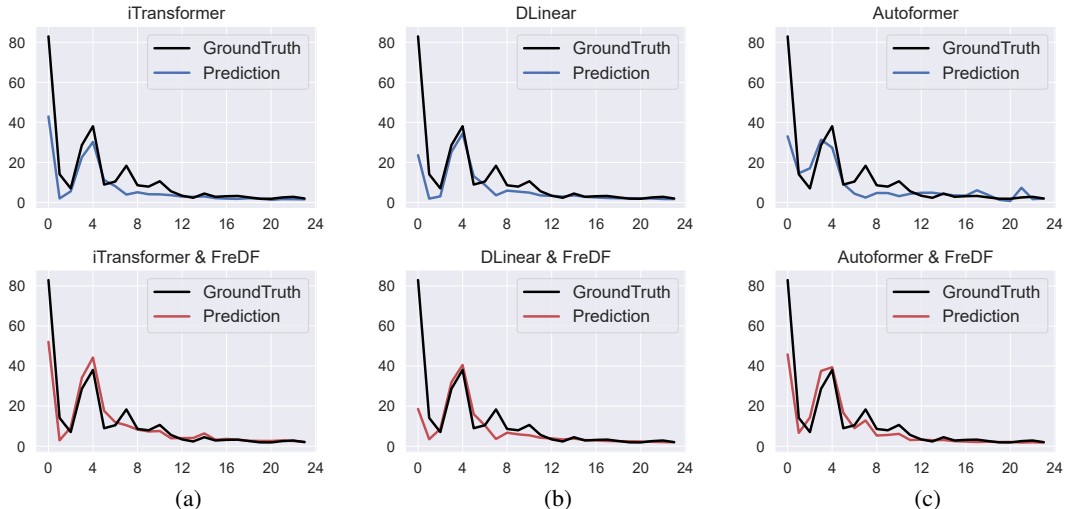

Figure 15: The spectrum of forecast sequences generated with DF and FreDF. The forecast length is set to 336 and the experiment is conducted on another snapshot of ETTm2. Only the first 24 frequencies of the spectrum are presented.

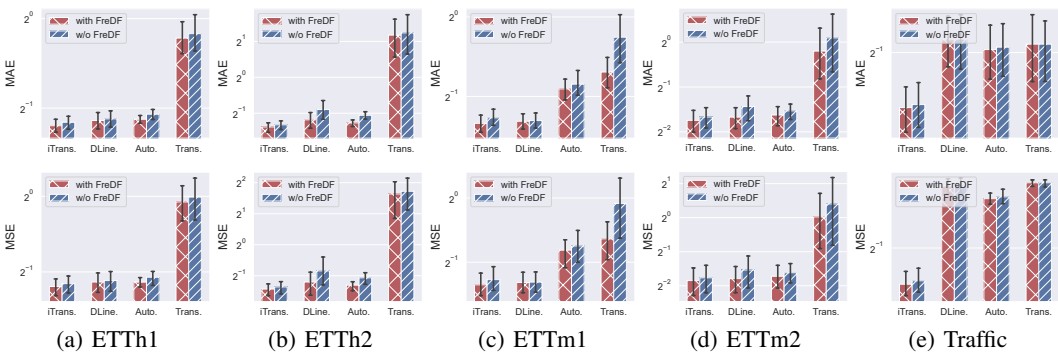

Figure 16: Performance of different forecast models with and without FreDF. The forecast errors are averaged over forecast lengths and the error bars represent 95% confidence intervals.

Table 8: Comparable results with DTW-based loss.

| Dataset | ETTm1 | | | | | | ETTh1 | | | | | |
|---|---|---|---|---|---|---|---|---|---|---|---|---|
| Models | **FreDF** | | Dilate | | DPP | | **FreDF** | | Dilate | | DPP | |
| Metrics | MSE | MAE | MSE | MAE | MSE | MAE | MSE | MAE | MSE | MAE | MSE | MAE |
| 96 | 0.324 | 0.362 | 0.498 | 0.443 | 0.631 | 0.495 | 0.382 | 0.400 | 0.790 | 0.567 | 0.815 | 0.577 |
| 192 | 0.373 | 0.385 | 0.993 | 0.625 | 0.975 | 0.617 | 0.430 | 0.427 | 0.950 | 0.643 | 0.916 | 0.633 |
| 336 | 0.402 | 0.404 | 0.946 | 0.628 | 0.945 | 0.626 | 0.474 | 0.451 | 0.978 | 0.663 | 0.986 | 0.660 |
| 720 | 0.469 | 0.444 | 0.999 | 0.652 | 1.079 | 0.678 | 0.463 | 0.462 | 0.922 | 0.654 | 0.898 | 0.649 |
| Avg | 0.392 | 0.399 | 0.859 | 0.587 | 0.907 | 0.604 | 0.437 | 0.435 | 0.910 | 0.632 | 0.904 | 0.630 |

### E.3 Hyperparameter sensitivity

In this section, we examine how adjusting the frequency loss weight $\alpha$ impacts the performance of FreDF across three models: iTransformer, Autoformer, and DLinear, with the results in Fig. 17, 18, and 19. We find that increasing $\alpha$ from 0 to 1 generally reduces forecast error across various datasets and forecast lengths, highlighting the benefits of a frequency domain learning approach. Notably, the minimum forecast error often occurs at $\alpha$ values close to 1, rather than at 1 itself; for instance, 0.8 is optimal for the ETTh1 dataset. This suggests that integrating supervisory signals from both time and frequency domains enhances forecasting performance. However, the improvement may be incremental compared to simply setting $\alpha = 1$.

### E.4 Comparison with DTW-based learning objectives

In this section, we compare FreDF with works that employ DTW as learning objectives to align the shape of the forecast sequence with the label sequence: Dilate (Le Guen & Thome, 2019) and DPP (Le Guen & Thome, 2020). Notably, these works do not handle the bias introduced by label autocorrelation, which makes them independent to the contribution of FreDF. To make a fair comparison, we integrated the official implementations of the loss functions into the iTransformer model. As shown in Table 8, FreDF significantly outperforms DTW-based methods across both datasets. This improvement stems from FreDF's unique ability to debias the learning objective, a capability that Dilate and DPP do not possess.

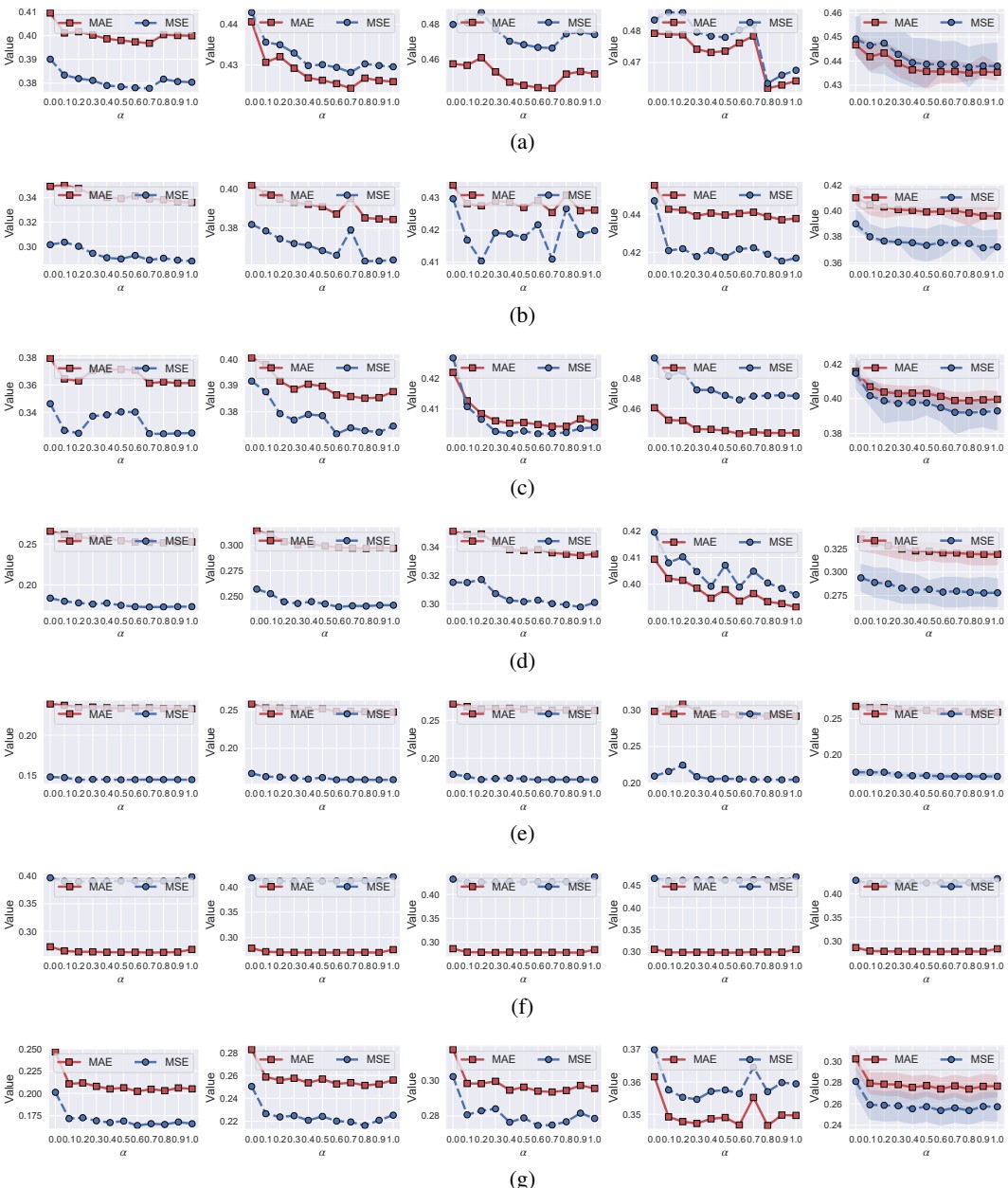

Figure 17: FreDF improves iTransformer performance given a wide range of frequency loss weight $\alpha$. These experiments are conducted on ETTh1 (a), ETTh2 (b), ETTm1 (c), ETTm2 (d), ECL (e), Traffic (f) and Weather (g) datasets. Different columns correspond to different forecast lengths (from left to right: 96, 192, 336, 720, and their average with shaded areas being 50% confidence intervals).

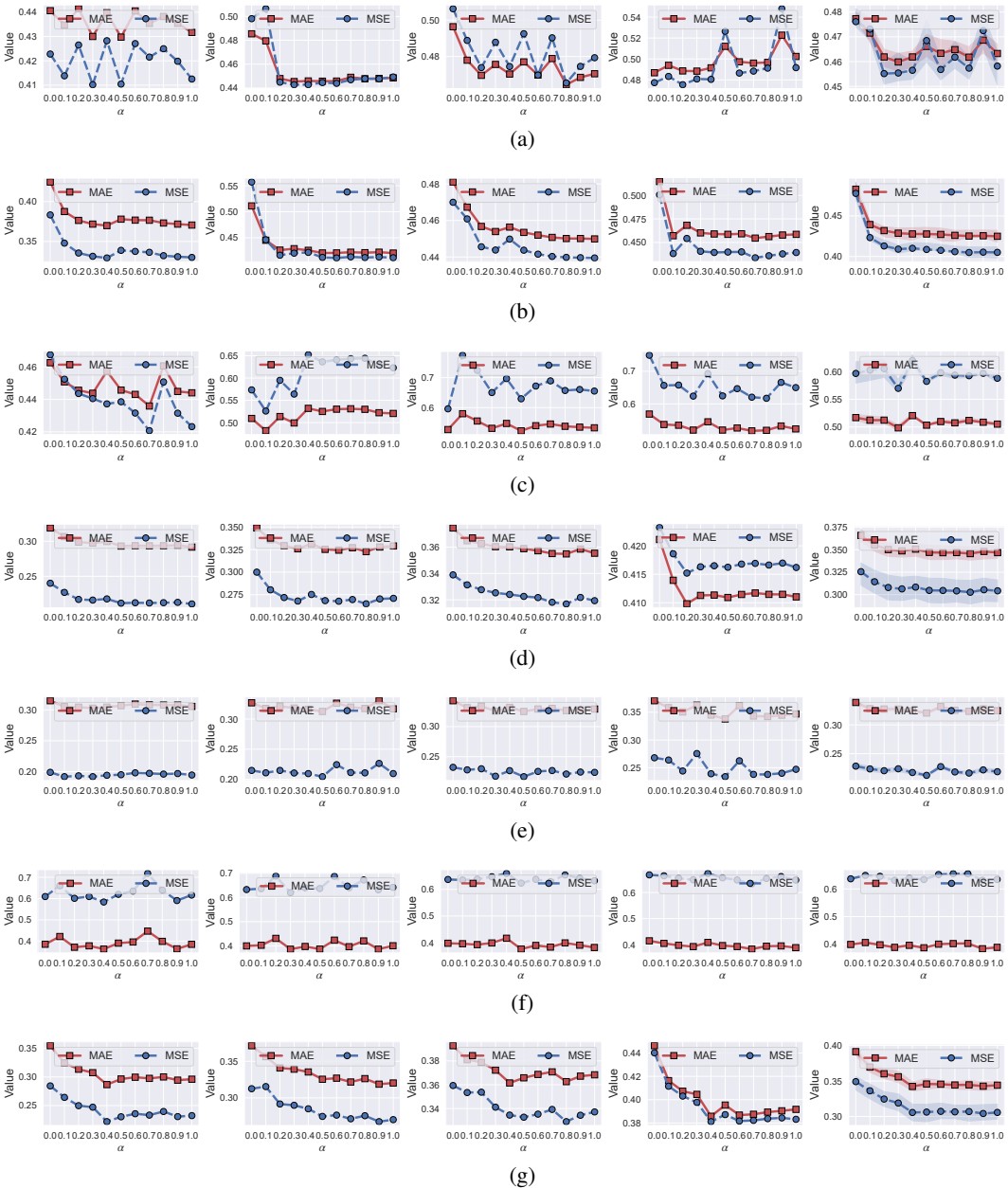

Figure 18: FreDF improves Autoformer performance given a wide range of frequency loss weight $\alpha$. These experiments are conducted on ETTh1 (a), ETTh2 (b), ETTm1 (c), ETTm2 (d), ECL (e), Traffic (f) and Weather (g) datasets. Different columns correspond to different forecast lengths (from left to right: 96, 192, 336, 720, and their average with shaded areas being 50% confidence intervals).

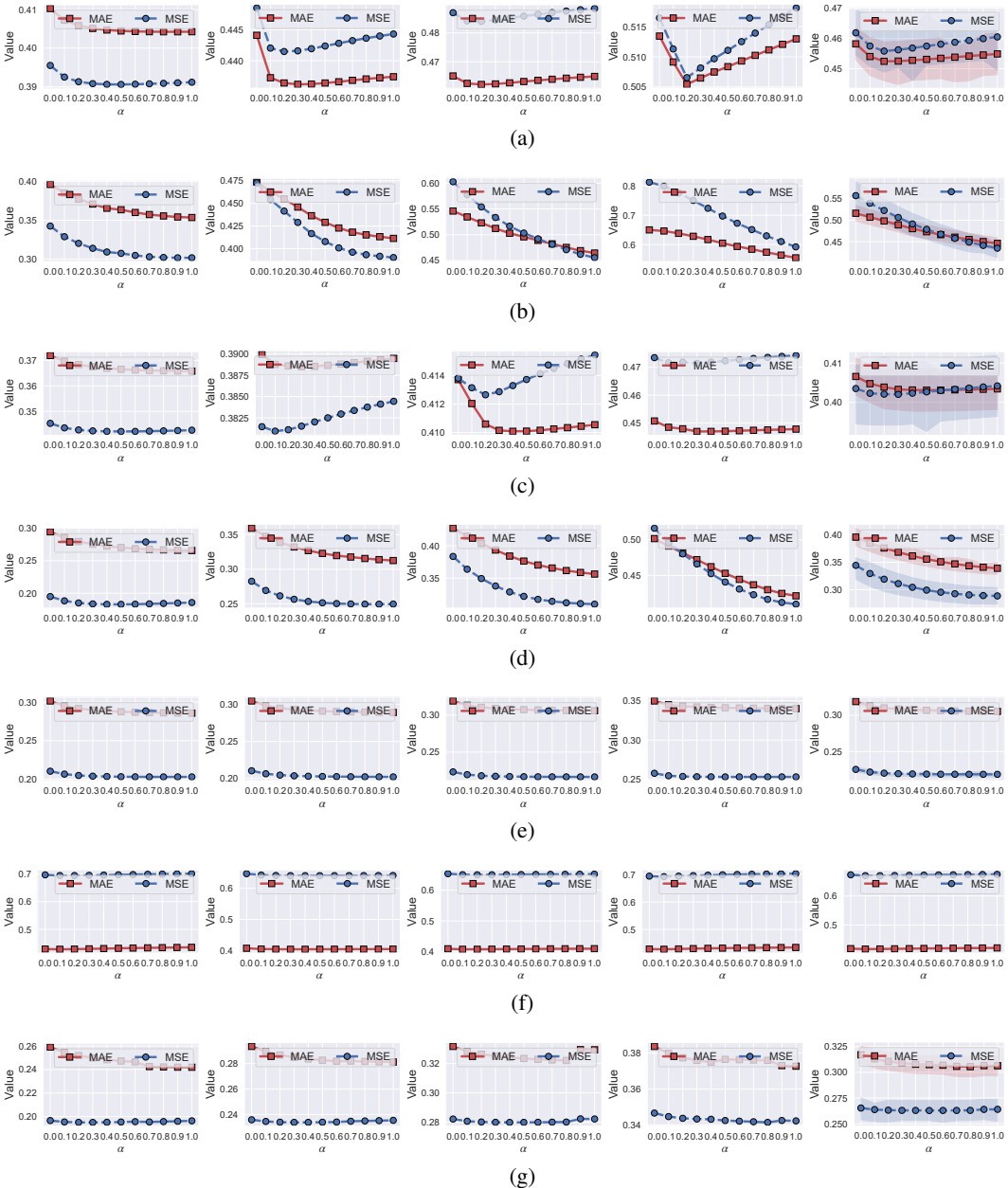

Figure 19: FreDF improves DLinear performance given a wide range of frequency loss weight $\alpha$. These experiments are conducted on ETTh1 (a), ETTh2 (b), ETTm1 (c), ETTm2 (d), ECL (e), Traffic (f) and Weather (g) datasets. Different columns correspond to different forecast lengths (from left to right: 96, 192, 336, 720, and their average with shaded areas being 50% confidence intervals).

