## A  CASE STUDY WITH PATCHTST AND VARYING INPUT LENGTH.

In this section, we focus on iTransformer (Liu et al., 2024) and PatchTST (Nie et al., 2023), highlighting the effectiveness of FreDF in enhancing their performance given varying input sequence lengths, to complement the fixed length of 96 used in the main text. According to Table 1, FreDF consistently improves the performance of both iTransformer and PatchTST across different input lengths. Notably, under our experimental conditions, PatchTST with H = 336 achieves results comparable to the original "PatchTST/42" results reported by Nie et al. (2023), while FreDF further reduced the MSE and MAE by 0.002, demonstrating its robustness across different input lengths.

Table 1: Varying input sequence length results on the Weather dataset.

| Models | | | FreDF | | iTransformer | | FreDF | | PatchTST | |
|---|---|---|---|---|---|---|---|---|---|---|
| Metrics | | | MSE | MAE | MSE | MAE | MSE | MAE | MSE | MAE |
| | 96 | 96 | 0.164 | 0.202 | 0.201 | 0.247 | 0.174 | 0.217 | 0.200 | 0.244 |
| | | 192 | 0.220 | 0.253 | 0.250 | 0.283 | 0.230 | 0.266 | 0.234 | 0.268 |
| | | 336 | 0.275 | 0.294 | 0.302 | 0.317 | 0.279 | 0.301 | 0.311 | 0.321 |
| | | 720 | 0.356 | 0.347 | 0.370 | 0.362 | 0.355 | 0.351 | 0.365 | 0.353 |
| | | Avg | 0.254 | 0.274 | 0.281 | 0.302 | 0.259 | 0.284 | 0.278 | 0.297 |
| | 192 | 96 | 0.164 | 0.207 | 0.184 | 0.235 | 0.158 | 0.205 | 0.167 | 0.213 |
| | | 192 | 0.211 | 0.250 | 0.236 | 0.277 | 0.200 | 0.241 | 0.204 | 0.244 |
| | | 336 | 0.262 | 0.290 | 0.268 | 0.296 | 0.259 | 0.287 | 0.266 | 0.291 |
| | | 720 | 0.341 | 0.343 | 0.342 | 0.345 | 0.330 | 0.334 | 0.333 | 0.337 |
| | | Avg | 0.244 | 0.272 | 0.258 | 0.288 | 0.237 | 0.267 | 0.242 | 0.271 |
| | 336 | 96 | 0.159 | 0.204 | 0.164 | 0.215 | 0.150 | 0.200 | 0.153 | 0.203 |
| | | 192 | 0.204 | 0.248 | 0.211 | 0.256 | 0.193 | 0.240 | 0.194 | 0.240 |
| | | 336 | 0.253 | 0.288 | 0.260 | 0.292 | 0.245 | 0.280 | 0.247 | 0.282 |
| | | 720 | 0.325 | 0.336 | 0.327 | 0.339 | 0.320 | 0.332 | 0.321 | 0.336 |
| | | Avg | 0.235 | 0.269 | 0.241 | 0.276 | 0.227 | 0.263 | 0.229 | 0.265 |
| | 720 | 96 | 0.164 | 0.215 | 0.172 | 0.228 | 0.144 | 0.194 | 0.191 | 0.246 |
| | | 192 | 0.209 | 0.257 | 0.218 | 0.265 | 0.190 | 0.242 | 0.192 | 0.241 |
| | | 336 | 0.251 | 0.291 | 0.273 | 0.306 | 0.243 | 0.283 | 0.241 | 0.285 |
| | | 720 | 0.318 | 0.342 | 0.340 | 0.353 | 0.310 | 0.330 | 0.311 | 0.331 |
| | | Avg | 0.236 | 0.276 | 0.251 | 0.288 | 0.222 | 0.262 | 0.234 | 0.276 |

(Input sequence length)

## B  RUNNING COST ANALYSIS

In this section, we analyze the running cost of FreDF. The core computation of FreDF involves calculating the FFT of both predicted and label sequences, followed by calculating their point-wise MAE loss. The overall complexity is dominated by the FFT operation, which operates at $\mathcal{O}(T \log T)$, where T is the label sequence length.

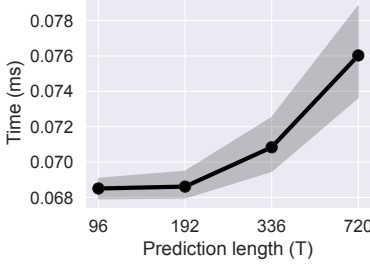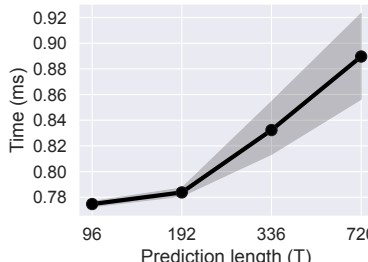

Figure 1: Running time in the forward pass (left panel) and backward pass (right panel), shown with dashed lines for the average and shaded areas for 99.9% confidence intervals.

Fig. 1 shows the empirical running costs of FreDF for varying sequence lengths in the training duration, involving the forward pass stage (FFT calculation) and the backward pass stage (frequency loss and gradient computation). Overall, for a label sequence with T < 720, FreDF adds approximately 1

Table 2: Experimental results ($\text{mean}_{\pm\text{std}}$) with varying seeds (2020-2024).

| Dataset | ETTh1 | | | | Weather | | | |
|---|---|---|---|---|---|---|---|---|
| Models | **FreDF** | | iTransformer | | **FreDF** | | iTransformer | |
| Metrics | MSE | MAE | MSE | MAE | MSE | MAE | MSE | MAE |
| 96 | $0.377_{\pm0.001}$ | $0.396_{\pm0.001}$ | $0.391_{\pm0.001}$ | $0.409_{\pm0.001}$ | $0.168_{\pm0.003}$ | $0.205_{\pm0.003}$ | $0.203_{\pm0.002}$ | $0.246_{\pm0.002}$ |
| 192 | $0.428_{\pm0.001}$ | $0.424_{\pm0.001}$ | $0.446_{\pm0.002}$ | $0.441_{\pm0.002}$ | $0.220_{\pm0.001}$ | $0.254_{\pm0.001}$ | $0.249_{\pm0.001}$ | $0.281_{\pm0.001}$ |
| 336 | $0.466_{\pm0.001}$ | $0.442_{\pm0.001}$ | $0.484_{\pm0.005}$ | $0.460_{\pm0.003}$ | $0.281_{\pm0.002}$ | $0.298_{\pm0.002}$ | $0.299_{\pm0.002}$ | $0.315_{\pm0.002}$ |
| 720 | $0.468_{\pm0.005}$ | $0.465_{\pm0.003}$ | $0.499_{\pm0.015}$ | $0.489_{\pm0.010}$ | $0.364_{\pm0.008}$ | $0.354_{\pm0.006}$ | $0.371_{\pm0.001}$ | $0.361_{\pm0.001}$ |
| Avg | $0.435_{\pm0.002}$ | $0.432_{\pm0.002}$ | $0.455_{\pm0.006}$ | $0.450_{\pm0.004}$ | $0.258_{\pm0.004}$ | $0.278_{\pm0.003}$ | $0.280_{\pm0.001}$ | $0.301_{\pm0.002}$ |

Table 3: Impact of aligning the amplitude and phase characteristics.

| Amp. | Pha. | ECL | | ETTm1 | | ETTh1 | |
|---|---|---|---|---|---|---|---|
| | | MSE | MAE | MSE | MAE | MSE | MAE |
| ✓ | ✗ | 0.3356 | 0.4060 | 0.5936 | 0.5169 | 0.7303 | 0.5968 |
| ✗ | ✓ | _0.1836_ | _0.2752_ | _0.4204_ | _0.4173_ | _0.4751_ | _0.4487_ |
| ✓ | ✓ | **0.1698** | **0.2594** | **0.3920** | **0.3989** | **0.4374** | **0.4351** |

ms to the overall training duration. Moreover, frequency loss computation is not required during inference. Therefore, FreDF does not hinder model efficiency in either training or inference stages.

## C  RANDOM SEED SENSITIVITY

In this section, we investigate the sensitivity of the results to the specification of random seeds. To this end, we report the mean and standard deviation of the results obtained from experiments using five random seeds (2020, 2021, 2022, 2023, 2024) in Table 2. We examine (1) iTransformer and (2) FreDF, which is applied to refine iTransformer. The results show minimal sensitivity to random seeds, with standard deviations below 0.005 in seven out of eight averaged cases.

## D  AMPLITUDE V.S. PHASE ALIGNMENT

In this section, we investigate the implementation of the frequency loss (3), with the results averaged over forecast lengths in Table 3. Specifically, minimizing the frequency loss (3) ensures that both amplitude and phase characteristics of the forecast match those of the actual label sequences in the frequency domain. In signal processing, both characteristics are fundamental for accurately representing signal dynamics, and we analyze their respective contributions. Overall, both characteristics are essential for FreDF's performance. Notably, phase alignment is particularly crucial; aligning amplitude characteristics without also aligning phase characteristics leads to subpar performance. This phenomenon is reasonable, as even minor deviations in phase characteristics can produce significant discrepancies in the time domain.

Table 4: Comparable results with baselines utilizing multiresolution trends.

| Dataset | ETTm1 | | | | | | | | ETTh1 | | | | | | | |
|---|---|---|---|---|---|---|---|---|---|---|---|---|---|---|---|---|
| Models | **FreDF** | | TimeMixer | | **FreDF** | | Scaleformer | | **FreDF** | | TimeMixer | | **FreDF** | | Scaleformer | |
| Metrics | MSE | MAE | MSE | MAE | MSE | MAE | MSE | MAE | MSE | MAE | MSE | MAE | MSE | MAE | MSE | MAE |
| 96 | 0.316 | 0.354 | 0.322 | 0.361 | 0.365 | 0.391 | 0.393 | 0.417 | 0.364 | 0.393 | 0.375 | 0.445 | 0.375 | 0.415 | 0.407 | 0.445 |
| 192 | 0.360 | 0.377 | 0.362 | 0.382 | 0.417 | 0.436 | 0.435 | 0.439 | 0.422 | 0.424 | 0.441 | 0.431 | 0.414 | 0.440 | 0.430 | 0.455 |
| 336 | 0.383 | 0.399 | 0.392 | 0.405 | 0.478 | 0.461 | 0.541 | 0.500 | 0.454 | 0.432 | 0.490 | 0.458 | 0.463 | 0.468 | 0.462 | 0.475 |
| 720 | 0.447 | 0.440 | 0.453 | 0.441 | 0.575 | 0.533 | 0.608 | 0.530 | 0.467 | 0.460 | 0.481 | 0.469 | 0.484 | 0.499 | 0.545 | 0.551 |
| Avg | 0.377 | 0.393 | 0.382 | 0.397 | 0.459 | 0.455 | 0.494 | 0.471 | 0.427 | 0.427 | 0.446 | 0.441 | 0.434 | 0.455 | 0.461 | 0.482 |

# E  COMPARISON WITH ADDITIONAL FORECAST ARCHITECTURES

In this section, we apply FreDF to two additional forecast architectures, namely TimeMixer (Wang et al., 2024d) and ScaleFormer (Shabani et al., 2022) to showcase the generality of FreDF. To ensure a fair comparison, we utilized their official repositories, downloading and configuring them according to their specified requirements. We modified their temporal MSE loss with the proposed loss in the FreDF. The loss strength parameters were fine-tuned on the validation set. As shown in Table 4, FreDF significantly enhances the performance of these architectures, demonstrating FreDF's ability to support and improve existing models. These improvements underscore the independent and complementary nature of FreDF's contributions.