# OpenReview forum: "FreDF: Learning to Forecast in the Frequency Domain"
_ICLR.cc/2025/Conference — ICLR 2025 Poster_

### Official Review · Reviewer_6T6T · 2024-11-01

**Soundness:** 3
**Presentation:** 3
**Contribution:** 3
**Rating:** 6
**Confidence:** 4

**Summary:**

This paper proposes to measure the distance between time series prediction outputs and target signals in the frequency domain.
As claimed, this allows to capture label autocorrelation in time series prediction tasks. Speciafically, in practice, distances in both time and frequency domains are utilized for training in a combined manner.

**Strengths:**

The presentation of the paper is clear and the idea is straightforward.

**Weaknesses:**

The proposed approach lacks novelty. Measuring time series structure (e.g., autocorrelation) in the frequency domain has been well-explored. Several prior works have investigated capturing time series characteristics or enhancing robustness in loss functions, including:
  1. Incorporating Fourier transformation in loss functions [1,2].
  1. Employing DTW-based loss to keep shape information of time series [3,4].
   1. Utilizing multiresolution trends during training [5,6].

These approaches are all relevant to this study. However, this paper does not adequately investigate, discuss, or compare these related methods.

[1] Henning Lange, et al. From Fourier to Koopman: Spectral Methods for Long-term Time Series Prediction. JMLR 2021.

[2] Xinyu Yuan and Yan Qiao. Diffusion-TS: Interpretable Diffusion for General Time Series Generation. In ICLR 2024

[3] Vincent Le Guen and Nicolas Thome. Shape and Time Distortion Loss for Training Deep Time Series Forecasting Models. In NeurIPS 2019.

[4] Vincent Le Guen and Nicolas Thome. Probabilistic time series forecasting with shape and temporal diversity. In NeurIPS 2020.

[5] Shiyu Wang, et al. TimeMixer: Decomposable Multiscale Mixing for Time Series Forecasting. In ICLR 2024.

[6] Amin Shabani, et al. Scaleformer: Iterative Multi-scale Refining Transformers for Time Series Forecasting. In ICLR 2023.

**Questions:**

See Weaknesses.

---

> ### Author Response · Authors · 2024-11-22
> **FreDF is distinctly differentiated from concerned references in multiple aspects, making unique contributions [1/4]**
>
> Thank you for taking the time to review our work and for appreciating our clarifications.
>
> We highly value your comments and have invested six working days to thoroughly investigate, discuss, and compare the related works [1-6]. This effort involved familiarizing ourselves with various environments and debugging the corresponding codes. We appreciate your understanding regarding our delayed response.
>
> **The conclusion is that FreDF is distinctly differentiated from these works in multiple aspects, making unique contributions.**
>
> ---
>
> #### [W1] The proposed approach lacks novelty. Measuring time series structure in the frequency domain has been well-explored. Several prior works have investigated capturing time series characteristics or enhancing robustness in loss functions. These approaches are all relevant to this study. However, this paper does not adequately `investigate, discuss, or compare` these related methods.
> **Response.** Thank you for your critical comment on related works.  To clearly address your concerns, we have structured our response as follows, aligning with your demand of `investigate, discuss, and compare`.
>
> | Section | Answers we provide |
> |---|---|
> | Investigation | What is the core research problem in these literatures? How do they technically address it? |
> | Discussion    | How does FreDF differ from them, in terms of research problems and technical contributions? |
> | Comparison    | Does FreDF outperform them under rigorously controlled experimental settings. |

---

> ### Author Response · Authors · 2024-11-22
> **FreDF is distinctly differentiated from concerned references in multiple aspects, making unique contributions [2/4]**
>
> #### [W1-1] Comparison with methods incorporating Fourier transformation in loss functions [1,2].
> - **Investigation.**
>   - Reference [1] achieves time series forecasting by identifying the parameters in a (linear) dynamic system model. FFT is used to **accelerate** the computation of a loss function tailored for dynamic systems. This approach is analogous to using Fourier analysis for solving ordinary differential equations (ODEs).
>   - Reference [2] focuses on the time series generation task based on diffusion models. A frequency loss is used to recover the original series $x_0$ to enhance interpretability and accuracy of the generated time series.
> - **Discussion.**
>   - **Research problems are different.** Fourier-DS [1] seeks to enhance the efficiency of training dynamic system models for time series forecasting by leveraging FFT to accelerate loss computations. Diffusion-TS [2] aims to advance diffusion models for conditional time-series generation by utilizing FFT-based frequency loss to improve interpretability and reconstruction fidelity. **In contrast, FreDF is specifically designed for forecasting tasks, employing FFT to eliminate the bias resulting from label autocorrelation, which is a new research problem.**
>   - **Implementations are different.** Fourier-DS requires the underlying model to be a dynamic system that performs recurrent inference for multi-step forecasts. Diffusion-TS relies on diffusion models, requiring training with score matching and inference through recurrent sampling. **In contrast, FreDF is model-agnostic, allowing seamless integration with a variety of forecasting models.** Additionally, it generates multi-step predictions simultaneously, thereby **avoiding the recurrent generation processes** in [1,2].
>
>   - **Most importantly, references [3,4] neither recognize nor address the bias introduced by label autocorrelation. FreDF bridges this gap by formally defining the bias (see Theorem 1) and mitigating it with theoretical guarantees. The identification and handling of this bias constitute the core contribution of FreDF, as highlighted in the title `Label correlation biases direct time-series forecast`.**
>
> |Literature|Task|Training paradigm | Inference paradigm | Model agnostic | Role of FFT | Recognizing label correlation | Formalizing bias | Handling bias |
> |---|---|---|---|---|---|---|---|---|
> Fourier-DS [1] | Forecast | Likelihood maximization | Recurrent inference | $\times$ | Acceleration | $\times$ | $\times$| $\times$ |
> Diffusion-TS [2] | Generation | Score matching | Recurrent sampling (algorithm 2 in [2]) | - | Interpretability and Reconstruction | $\times$ | $\times$|$\times$ |
> FreDF (Ours) | Forecast | Likelihood maximization | Multitask inference | $\checkmark$ | Debiasing | $\checkmark$ | $\checkmark$ | $\checkmark$|
>
> - **Comparison.**
>   - **Setup.** To ensure a fair comparison, we utilized the official repositories of [1,2], downloading and configuring them according to their specified requirements. We processed the datasets using the unified protocol of TSLib [7] and adjusted the forecasting horizon lengths to perform long-term forecasts.
>   - **Observation.** As presented in the table below, FreDF significantly outperforms the baselines [1,2] across both datasets. This outcome aligns with our expectations based on prior discussions. Fourier-DS [1] is restricted to a (linear) dynamic system model, limiting its ability to capture complex temporal patterns compared to advanced deep models like iTransformer, which leads to suboptimal performance. Diffusion-TS [2] is not specifically designed for time-series forecasting; although it can be adapted for forecasting via conditional generation, the inherent diversity of diffusion model outputs prove to adversely affect accuracy metrics [8,9]. In contrast, FreDF supports modern forecasting models without relying on generative paradigms, effectively debiasing the learning objective and achieving superior performance.
>
> | Dataset| **ETTm1** |||||| **ETTh1**||||| |
> |---------|---------|-----|------|------|------|------|------|------|------|------|------|------|
> | Models | **FreDF** || Fourier-DS [1]|| Diffusion-TS [2] || **FreDF**|| Fourier-DS [1]|| Diffusion-TS [2]||
> | Metrics| MSE| MAE| MSE| MAE| MSE| MAE| MSE| MAE| MSE| MAE| MSE| MAE|
> | **96** | 0.324| 0.362| 1.685| 0.984| 0.731| 0.637| 0.382| 0.400| 1.651| 0.996| 1.202| 0.882|
> | **192**| 0.373| 0.385| 1.687| 0.984| 0.745| 0.640| 0.430| 0.427| 1.669| 1.002| 1.133| 0.849|
> | **336**| 0.402| 0.404| 1.691| 0.984| 0.812| 0.665| 0.474| 0.451| 1.697| 1.011| 1.290| 0.918|
> | **720**| 0.469| 0.444| 1.709| 0.989| 0.974| 0.756| 0.463| 0.462| 1.776| 1.036| 1.124| 0.846|
> | **Avg**| 0.392| 0.399| 1.693| 0.985| 0.816| 0.674| 0.437| 0.435| 1.698| 1.011| 1.187| 0.886|

---

> ### Author Response · Authors · 2024-11-22
> **FreDF is distinctly differentiated from concerned references in multiple aspects, making unique contributions [3/4]**
>
> #### [W1-2] Comparison with DTW-based loss to keep shape information of time series [3,4].
> - **Investigation.**
>   - Reference [3] focuses on point forecasting for non-stationary signals. It proposes DILATE, which aligns the shape of the forecast sequence with the label sequence using a differentiable Dynamic Time Warping (DTW) loss
>   - Reference [4] focuses on the probabilistic forecast task for non-stationary time series. It proposes STRIPE which captures structured diversity based on shape and temporal features using DTW.
> - **Discussion.**
>   - **Research problems are different.** References [3-4] focuses on forecast tasks for non-stationary signals, specifically targeting **the alignment of forecast and label sequence shapes**, which improves performance in an **intuitive manner**. In contrast, FreDF is engineered for general forecasting tasks, specifically targeting **the elimination of bias within the learning objective**, which improves performance with **theoretical guarantees**.
>   - **Implementations are different.** While references [3] and [4] rely on modified DTW loss, FreDF utilizes frequency-based loss, which operates independently of DTW loss. Furthermore, FreDF is adaptable to various loss functions, such as Legendre and Chebyshev losses (see Section 4.4), highlighting its flexibility and difference from [3-4].
>   - **Most importantly, references [3,4] neither recognize nor address the bias introduced by label autocorrelation. FreDF bridges this gap by formally defining the bias (see Theorem 1) and mitigating it with theoretical guarantees. The identification and handling of this bias constitute the core contribution of FreDF, as highlighted in the title `Label correlation biases direct time-series forecast`.**
>
>
> Literature | Task | Motivation | Using frequency loss | Theoretical guarantee | Formalizing bias | Handling bias |
> |---|---|---|---|---|---|---|
> DILATE [3] | Point Forecast | Align shape | $\times$ | $\times$ |  $\times$ | $\times$ |
> STRIPE [4] | Probabilistic Forecast | Align shape | $\times$ | $\times$ | $\times$| $\times$ |
> FreDF (Ours) | Point Forecast | Debias | $\checkmark$ | $\checkmark$ | $\checkmark$ | $\checkmark$ |
>
> - **Comparison.**
>   - **Setup.** To ensure a fair comparison, we integrated the official implementations of the loss functions [3,4] into the iTransformer model within the TSLib framework [7]. We adhered to the forecasting and input length settings specified by TSLib to perform long-term forecasts. The loss strength parameters were fine-tuned on the validation set within the range [0.1, 0.3, 0.5, 0.7, 1].
>   - **Observation.** As shown in the table below, FreDF significantly outperforms the baseline methods [1,2] across both datasets. This result aligns with our expectations and the observations reported in References [3,4]. Specifically, as highlighted in Table 1 and the first paragraph of Reference [3]: `DILATE is equivalent to MSE when evaluated on MSE on 3/6 experiments.` Indeed, for MLP-like models incorporating DILATE, forecast accuracy measured by MSE decreases. In contrast, FreDF significantly outperforms DTW-based methods across both datasets. This improvement stems from FreDF’s unique ability to debias the learning objective, a capability that DILATE and DPP do not possess.
>
> | Dataset| **ETTm1** ||||||**ETTh1**||||||
> |--|--|--|--|--|--|--|--|--|--|--|--|--|
> | Models         | **FreDF (Ours)** || Dilate [3]        || DPP  [4]          || **FreDF (Ours)** || Dilate [3]        || DPP  [4]          ||
> | Metrics        | MSE     | MAE     | MSE    | MAE    | MSE    | MAE    | MSE     | MAE     | MSE    | MAE    | MSE    | MAE    |
> | **96**         | 0.324   | 0.362   | 0.498   | 0.443   | 0.631   | 0.495   | 0.382   | 0.400   | 0.790   | 0.567   | 0.815   | 0.577   |
> | **192**        | 0.373   | 0.385   | 0.993   | 0.625   | 0.975   | 0.617   | 0.430   | 0.427   | 0.950   | 0.643   | 0.916   | 0.633   |
> | **336**        | 0.402   | 0.404   | 0.946   | 0.628   | 0.945   | 0.626   | 0.474   | 0.451   | 0.978   | 0.663   | 0.986   | 0.660   |
> | **720**        | 0.469   | 0.444   | 0.999   | 0.652   | 1.079   | 0.678   | 0.463   | 0.462   | 0.922   | 0.654   | 0.898   | 0.649   |
> | **Avg**        | 0.392   | 0.399   | 0.859   | 0.587   | 0.907   | 0.604   | 0.437   | 0.435   | 0.910   | 0.632   | 0.904   | 0.630   |

---

> ### Author Response · Authors · 2024-11-22
> **FreDF is distinctly differentiated from concerned references in multiple aspects, making unique contributions [4/4]**
>
> #### [W1-3] Comparison with methods utilizing multiresolution trends during training [5,6].
> - **Investigation.**
>   - Reference [5] focuses on architecture development. It proposes Timemixer that decomposes multiscale series and successively aggregates the microscopic and macroscopic information.
>   - Reference [6] also focuses on architecture development. It proposes Scaleformer that refines a forecasted time series at multiple scales with shared weights.
>
> - **Discussion.**
>   - **Research problems are different.** References [5-6] targets utilizing the multi-scale information to enhance performance. In contrast, FreDF targets **the elimination of bias within the learning objective****.
>   - **Implementations are different.** References [5-6] develop new network architectures. In contrast, FreDF develops new loss function, which supports diverse network architectures (see results in Section 4.4).
>   - **Most importantly, references [5-6] neither recognize nor address the bias introduced by label autocorrelation. FreDF bridges this gap by formally defining the bias (see Theorem 1) and mitigating it with theoretical guarantees. The identification and handling of this bias constitute the core contribution of FreDF, as highlighted in the title `Label correlation biases direct time-series forecast`.**
>
> |Literature | Contribution | Target |
> |--|--|--|
> |TimeMixer [5] | Architecture | Incorporating multi-scale information |
> |ScaleFormer [6] | Architecture | Incorporating multi-scale information |
> |FreDF (Ours) | Loss function|Debiasing learning objective |
>
> - **Comparison.**
>   - **Setup.** To ensure a fair comparison, we utilized the official repositories of [5-6], downloading and configuring them according to their specified requirements. We modified their temporal MSE loss with the proposed loss in the FreDF. We processed the datasets using the unified protocol of TSLib [7] and adjusted the forecasting horizon lengths to perform long-term forecasts. The loss strength parameters were fine-tuned on the validation set within the range [0.1, 0.3, 0.5, 0.7, 1].
>   - **Observation.** As shown in the table below, FreDF significantly enhances the performance of the architectures [5,6], demonstrating FreDF’s ability to support and improve existing models. **These improvements underscore the independent and complementary nature of FreDF’s contributions.**
>
> | Dataset| ETTm1|||||||| ETTh1||||||||
> |----------------|---------|---------|---------|---------|---------|----------|---------|---------|---------|---------|---------|---------|---------|---------|---------|---------|
> | Models | **TimeMixer+FreDF** || TimeMixer|| **Scaleformer+FreDF** || Scaleformer|| **TimeMixer+FreDF** || TimeMixer|| **Scaleformer+FreDF** || Scaleformer||
> | Metrics| MSE | MAE | MSE| MAE| MSE| MAE | MSE| MAE| MSE | MAE| MSE| MAE| MSE | MAE| MSE| MAE|
> | **96**| 0.316 | 0.354 | 0.322 | 0.361 | 0.365 | 0.391| 0.393 | 0.417 | 0.364 | 0.393 | 0.375 | 0.445 | 0.375 | 0.415 | 0.407 | 0.445 |
> | **192**| 0.360 | 0.377 | 0.362 | 0.382 | 0.417 | 0.436| 0.435 | 0.439 | 0.422 | 0.424 | 0.441 | 0.431 | 0.414 | 0.440 | 0.430 | 0.455 |
> | **336**| 0.383 | 0.399 | 0.392 | 0.405 | 0.478 | 0.461| 0.541 | 0.500 | 0.454 | 0.432 | 0.490 | 0.458 | 0.463 | 0.468 | 0.462 | 0.475 |
> | **720**| 0.447 | 0.440 | 0.453 | 0.441 | 0.575 | 0.533| 0.608 | 0.530 | 0.467 | 0.460 | 0.481 | 0.469 | 0.484 | 0.499 | 0.545 | 0.551 |
> | **Avg**| 0.377 | 0.393 | 0.382 | 0.397 | 0.459 | 0.455| 0.494 | 0.471 | 0.427 | 0.427 | 0.446 | 0.441 | 0.434 | 0.455 | 0.461 | 0.482   |
>
> **Revision actions.**
> - We have discussed reference [5-6] in the lines 91-93, and explained the distinction of FreDF from [1-4] in the lines 158-161.
> - We have involved the empirical comparison results with [3-4] in Table 11 to highlight the advantage of FreDF.
> - We have involved the empirical comparison results with [5-6] in Table 12 to highlight the generality of FreDF.
>
> ---
> Reference
>
> [1] Henning Lange, et al. From Fourier to Koopman: Spectral Methods for Long-term Time Series Prediction. JMLR 2021.
>
> [2] Xinyu Yuan and Yan Qiao. Diffusion-TS: Interpretable Diffusion for General Time Series Generation. In ICLR 2024
>
> [3] Vincent Le Guen and Nicolas Thome. Shape and Time Distortion Loss for Training Deep Time Series Forecasting Models. In NeurIPS 2019.
>
> [4] Vincent Le Guen and Nicolas Thome. Probabilistic time series forecasting with shape and temporal diversity. In NeurIPS 2020.
>
> [5] Shiyu Wang, et al. TimeMixer: Decomposable Multiscale Mixing for Time Series Forecasting. In ICLR 2024.
>
> [6] Amin Shabani, et al. Scaleformer: Iterative Multi-scale Refining Transformers for Time Series Forecasting. In ICLR 2023.
>
> [7] Wang, Yuxuan, et al. "Deep time series models: A comprehensive survey and benchmark." arXiv preprint arXiv:2407.13278 (2024).
>
> [8] Chen, Zhichao, et al. "Rethinking the Diffusion Models for Missing Data Imputation: A Gradient Flow Perspective." In NeurIPS 2024.

---

> ### Author Response · Authors · 2024-11-25
>
> Dear Reviewer 6T6T,
>
> As the discussion deadline approaches, we would like to inquire whether our responses have adequately addressed your concerns. As the remaining opposing reviewer, your feedback is invaluable to us. During the rebuttal stage, our team has dedicated six working days to conduct additional experiments comparing all the referenced methods.
>
> To date, we have thoroughly `investigated, discussed, and compared` these methods in both our response letter and the revised manuscript.
>
> Should you have any further comments or questions, we would greatly appreciate the opportunity to address them.
>
> Thank you in advance,
>
> 13602 Authors

---

> ### Comment · Reviewer_6T6T · 2024-11-25
>
> The author has thoroughly addressed the distinctions between the relevant methods discussed and effectively emphasized their contributions. I am satisfied with the revisions made to the manuscript and have updated my rating accordingly.

---

> > ### Author Response · Authors · 2024-11-27
> > **Thank you for your support!**
> >
> > Thank you very much for your patience to read our comprehensive response and generosity to increase the rating -- We truly appreciate it!

---

### Official Review · Reviewer_WM7u · 2024-11-02

**Soundness:** 3
**Presentation:** 3
**Contribution:** 3
**Rating:** 8
**Confidence:** 4

**Summary:**

The paper presents a new Frequency-enhanced Direct Forecast method for time series predictions that aims to improve the predictions generated by direct forecast (DF) models by training the models using a combination of standard MSE errors and errors defined in the FFT transformed space. The new approach is motivated by the fact that the MSE errors for direct forecast (DF) models are biased and do not properly account for correlations in predicted sequences, while errors on the FFT on the predicted sequences may be more robust to such dependencies.  The experiments on multiple datasets and multiple baseline DF models demonstrate the improved performance of the new method over baselines.

**Strengths:**

Originality: The idea of augmenting an objective function with a loss in alternate bases that is more robust to value dependences in prediction sequences, in this case Fourier bases, is novel and interesting.

Significance: The extension applies to a subclass of direct forecasting models that predict components of future sequences independently, ignoring dependences that may exist among them.  This makes the method potentially applicable to a variety of SOTA time series models.

Quality: The authors attempt to analyze the problem of using simple additive error functions for predicting sequences and its fixes via errors in the FFT transformed space both theoretically and experimentally. Extensive experimentation across multiple datasets and tasks show the merits of the proposed approach compared to SOTA baselines.

Other strengths: Exploration of properties and extension of the proposed framework and its benefits, such as, different prediction length, ablation models,  possible new transformations and errors defined on these transformations.

Code: Authors provide the code for reproducing the experiments.

**Weaknesses:**

Limited analysis of a bias in Theorem 1: Theorem 1 considers only univariate sequences, cross-correlation terms are not accounted for in the Bias formula. The sequence has dimensionality D.

Notational inconsistency and formula errors in the paper:
- The use of 'L' is overloaded, it denotes both the length and inputs of the input sequence.
- Equation 1 has a mistake. Y and Y_hat should be compared on the same indexes.
- Definition 3.2, uses 'j' instead of “i”

The main results in the paper do not report performance over distinct random seeds. Given this is an experimental study, reporting the results on multiple distinct random seed could help to understand its sensitivity.

The short-term forecasting task results in the appendix report only qualitative time spans. It would be good to specify the results in terms of forecasting lengths, similar to the Table 1 for long-term forecasting.

**Questions:**

Are there any settings where defining the errors by transferring the predictions to the frequency domain is detrimental to the prediction accuracy?

---

> ### Author Response · Authors · 2024-11-22
>
> Thank you very much for your positive comments and appreciation of our **novelty, generality and empirical performance**. Below are our responses to the specific concerns and queries.
>
> ---
>
> #### **[W1] Limited analysis of a bias in Theorem 1. Theorem 1 considers only univariate sequences, cross-correlation terms are not accounted for in the Bias formula.**
> **Response.** We understand that Theorem 1 can be extended to multivariate cases. We have made theoretical and empirical efforts as follows.
> - **Firstly, we have added Corollary B.3 that extends Theorem 1 to multivariate case.** It immediately follows from Theorem 1 by denoting the multivariate label sequence $Z$ as an augmented univariate sequence. It demonstrates that the label correlation between different time steps and covariates leads to bias. The theorem is formulated as follows.
> > Given a input sequence $L$ and a multivariate label sequence $Y\in\mathbb{R}^\mathrm{T\times D}$, suppose $Z\in\mathbb{R}^\mathrm{T\times D}$ be the flattened version of $Y$ obtained by concatenating the rows, the learning objective of the DF paradigm is biased against the practical NLL, expressed as:
> $$
>     \mathrm{Bias} = \sum\_{i=1}\^\mathrm{T\times D} \frac{1}{2\sigma^2} (Z_i - \hat{Z}\_i)\^2 - \sum\_{i=1}\^\mathrm{T\times D} \frac{1}{2\sigma\^2 (1 - \rho_i\^2)} \left(Z\_i - \left(\hat{Z}\_i + \sum\_{j=1}\^{i-1} \rho\_{ij} (Z_j - \hat{Z}\_j)\right)\right)\^2,
> $$
> where $\hat{Z}\_i$ indicates the prediction of $Z\_i$, $\rho\_{ij}$ denotes the partial correlation between $Z\_i$ and $Z\_j$, $\rho\_i^2 = \sum\_{j=1}\^{i-1} \rho\_{ij}^2.$
>
> - **Secondly, to counteract the bias of cross-correlation terms, we perform FFT along the covariate dimension (FreDF-D). The results and observations are presented below.**
>   - In general, FreDF-D brings similar performance gain with FreDF, which showcases **the efficacy of FreDF-D to counteract the bias of cross-correlation terms.** In particular, FreDF-T slightly outperforms FreDF-D, which underscores the relative importance of auto-correlation in the label sequence. Finally,
>   - Notably, a strategic approach is viewing the multivariate sequence as an image, performing 2-dimensional FFT on both time and covariate axes (FreDF-2), **which accommodates the correlations between both time steps and covariates simultaneously** and further improves performance.
>
> | Data           | ETTh1|||       | ETTm1|||        | ECL |||          |
> |------------------|------------|------------|------------|------------|------------|------------|---|---|---|---|---|---|
> | Metric  | MSE        | ΔMSE | MAE  | ΔMAE   | MSE        | Δ MSE          | MAE        | Δ MAE          | MSE        | Δ MSE          | MAE        | Δ MAE          |
> | iTransformer     | 0.449      | -| 0.447| -           | 0.415      | -| 0.416| -| 0.176      | -           | 0.267      | -|
> | + FreDF        | 0.437      | $\downarrow$2.63%     | 0.435      | $\downarrow$2.62%     | 0.392| $\downarrow$5.49%     | 0.399      | $\downarrow$4.01%| 0.170      | $\downarrow$3.41%     | 0.259      | $\downarrow$2.77%     |
> | + FreDF-D| 0.445      | $\downarrow$0.92%| 0.440      | $\downarrow$1.42%     | 0.395      | $\downarrow$4.77%     | 0.398      | $\downarrow$4.33%     | 0.171| $\downarrow$2.51%     | 0.260| $\downarrow$2.52%|
> | + FreDF-2| 0.432| $\downarrow$3.94%| 0.431      | $\downarrow$3.57%| 0.392| $\downarrow$5.60%     | 0.399      | $\downarrow$4.05%| 0.166| $\downarrow$5.32%     | 0.256| $\downarrow$4.20%|
>
> #### **[W2] Notational inconsistency and formula errors in the paper: The use of 'L' is overloaded; Y and Y_hat in Eq.(1) should be compared on the same indexes; Definition 3.2, uses 'j' instead of 'i'.**
> **Response.** Thank you very much for your meticulous comment sincerely! **We have revised these typological errors in the revised manuscript.**
> - We have replaced $\mathrm{L}$ with $\mathrm{H}$ to denote the input length
> - We have revised Eq.(1) as $\mathcal{L}^{(\mathrm{tmp})} := \sum_{t=1}^\mathrm{T} || Y_t - \hat{Y}_t ||_2^2.$
> - We have revised Definition 3.2, as well as Theorem B.4, where we use $j$ to denote imaginary unit.
>
> #### **[W3] The main results in the paper do not report performance over distinct random seeds.**
> **Response.** We acknowledge the importance of evaluating performance variability across different random seeds.
> - **Additional experiment.** We report the mean and standard deviation of the results by conducting experiments using 5 random seeds (2020, 2021, 2022, 2023, 2024). We investigate (1) iTransformer, which is the best baseline model; (2) FreDF, which is applied to refine iTransformer.
> - **Result analysis.** The results indicate minimal performance variation, with standard deviations below 0.005 in 7 out of 8 Avg cases. This showcases the insensitivity of models in time series forecast to different random seed specifications.
> - **Revision.**  We have added a new section (see Appendix E.3) that include these analysis and results.

---

> ### Author Response · Authors · 2024-11-22
>
> **Results with random seeds on the ETTh1 dataset.**
> | Metrics | MSE (FreDF)         | MAE (FreDF)         | MSE (iTransformer) | MAE (iTransformer) |
> |---------|---------------------|---------------------|---------------------|---------------------|
> | 96      | 0.377 ± 0.001       | 0.396 ± 0.001       | 0.391 ± 0.001       | 0.409 ± 0.001       |
> | 192     | 0.428 ± 0.001       | 0.424 ± 0.001       | 0.446 ± 0.002       | 0.441 ± 0.002       |
> | 336     | 0.466 ± 0.001       | 0.442 ± 0.001       | 0.484 ± 0.005       | 0.460 ± 0.003       |
> | 720     | 0.468 ± 0.005       | 0.465 ± 0.003       | 0.499 ± 0.015       | 0.489 ± 0.010       |
> | **Avg** | 0.435 ± 0.002       | 0.432 ± 0.002       | 0.455 ± 0.006       | 0.450 ± 0.004       |
>
> **Results with random seeds on the Weather dataset.**
>
> | Metrics | MSE (FreDF)         | MAE (FreDF)         | MSE (iTransformer) | MAE (iTransformer) |
> |---------|---------------------|---------------------|---------------------|---------------------|
> | 96      | 0.168 ± 0.003       | 0.205 ± 0.003       | 0.203 ± 0.002       | 0.246 ± 0.002       |
> | 192     | 0.220 ± 0.001       | 0.254 ± 0.001       | 0.249 ± 0.001       | 0.281 ± 0.001       |
> | 336     | 0.281 ± 0.002       | 0.298 ± 0.002       | 0.299 ± 0.002       | 0.315 ± 0.002       |
> | 720     | 0.364 ± 0.008       | 0.354 ± 0.006       | 0.371 ± 0.001       | 0.361 ± 0.001       |
> | **Avg** | 0.258 ± 0.004       | 0.278 ± 0.003       | 0.280 ± 0.001       | 0.301 ± 0.002       |
>
>
> #### **[W4] The short-term forecasting task results in the appendix report only qualitative time spans. It would be good to specify the results in terms of forecasting lengths, similar to the Table 1 for long-term forecasting.**
> **Response.** Thank you very much for your kind suggestion in this regard.
>
> - In our experiments, each qualitative term corresponds to specific subsets within the M4 dataset, each with distinct input and forecasting lengths as detailed in the table below. These configurations follow the standard setups defined by the Time Series Library [1].
>
> - Since in the standard protocol [1], different trials in short-time forecast experiment indicates different subset of dataset, input length and label length, we used the qualitative term to specify the results, in order to encapsulate these variations concisely and align with the standard setting.
>
> | Dataset | Input Length | Label Length |
> |---|---|---|
> | M4-Yearly | 12 | 6 |
> | M4-Quarterly | 16 | 8 |
> | M4-Monthly | 36 | 18 |
> | M4-Weekly | 26 | 13 |
> | M4-Daily | 28 | 14 |
> | M4-Hourly | 96 | 48 |
>
> [1] Wang, Yuxuan, et al. "Deep time series models: A comprehensive survey and benchmark." arXiv preprint arXiv:2407.13278 (2024).
>
> #### **[Q1] Are there any settings where defining the errors by transferring the predictions to the frequency domain is detrimental to the prediction accuracy?**
> **Response.** Yes, there are specific scenarios where FFT can adversely affect prediction accuracy, particularly when dealing with irregular time series (although it is not the focus of this paper).
> - Irregular time series are characterized by non-uniform sampling intervals (e.g., observations recorded at varying frequencies such as days versus hours). FFT inherently assumes uniform sampling intervals, which means it cannot accurately capture the underlying frequency components of irregularly sampled data. Applying FFT directly to such time series disregards their temporal irregularities, leading to distorted frequency representations and, consequently, suboptimal prediction performance.
> - To address this limitation, a promising approach is extending FreDF by involving the Non-uniform Discrete Fourier Transform (NDFT). Unlike FFT, NDFT is designed to handle data sampled at irregular intervals. The NDFT is defined as: $X_k = \sum_{n}^{N-1} x_n e^{-2\pi i p_n f_k}$, where $k$ (ranging from $0$ to  $N-1$ ) denotes the frequency index, $p_0, \dots, p_{N-1}$  are the scaled time stamps of the samples, and  $f_0, \dots, f_{N-1}$  are the corresponding frequencies. By leveraging NDFT, we can compute spectral distances for irregular time series, thereby extending FreDF to irregular settings.
> -  We acknowledge that handling irregular time series is a limitation of the current FreDF implementation and represents a plausible direction for future efforts. It has been introduced in the final paragraph of our conclusion. Thank you again for your fruitful advice.

---

> > ### Comment · Reviewer_WM7u · 2024-11-28
> > **Comment of authors' response**
> >
> > I am satisfied with the response of the authors to the points and questions raised in my review and I am updating my rating of the paper.

---

> ### Author Response · Authors · 2024-11-29
> **Thank you for your continued and increased support!**
>
> Thank you very much for reviewing our response and for your dedication in adjusting the evaluation score! We truly appreciate your increased support.

---

### Official Review · Reviewer_3nNT · 2024-11-02

**Soundness:** 3
**Presentation:** 3
**Contribution:** 3
**Rating:** 6
**Confidence:** 3

**Summary:**

Considering that in current time series prediction, DF only reuses general multi task learning methods without considering the dependency relationships  between labels. Especially, even if there is no correlation between labels, the above loss function can be used to train a multitasking model. When training the model, DF attempts to minimize the error between the predicted label sequence and the true label sequence; This assumes that the label sequence is conditionally independent between different time steps, thereby ignoring the correlation between each time step within the label sequence. So the author uses fast Fourier transform in the frequency domain to transform the data from a temporal perspective to a frequency perspective, in order to suppress autocorrelation. And the theoretical derivation of using frequency domain conversion to suppress time step correlation is given. A loss function combining time domain and frequency domain is designed, and the effectiveness of the design method is demonstrated by experimental results.

**Strengths:**

1. Solving the problem of autocorrelation in label sequences: The FreDF method proposed in this paper effectively bypasses the autocorrelation problem in label sequences by predicting in the frequency domain. This is a common but not fully addressed problem in existing direct prediction models.
2. Compatible with multiple prediction models: The FreDF method is not only suitable for existing state-of-the-art methods such as iTransformer, but can also be compatible with multiple prediction models. This compatibility makes it widely applicable in different prediction tasks.
3. Significant improvement in predictive performance: Experimental results show that the FreDF method is significantly superior to existing methods in multi-step prediction tasks. This indicates that the method has high accuracy and reliability in processing complex time series data.
4. Innovative applications of frequency domain analysis: By introducing frequency domain analysis into time series prediction, the FreDF method provides a new perspective to address autocorrelation issues in time series. This innovative application not only improves predictive performance, but also provides new directions for future research.

**Weaknesses:**

1. Complexity of frequency domain conversion: Although frequency domain analysis can bypass the autocorrelation problem in label sequences, frequency domain conversion itself may increase computational complexity and time cost. This may become a bottleneck when dealing with large-scale datasets.
2. Model generalization ability: Although FreDF performed well in experiments, its generalization ability in different fields and datasets still needs further validation. Especially in practical applications, the model may need to be adjusted and optimized for specific tasks.
3. Dependence on frequency domain features: The FreDF method relies on the extraction and utilization of frequency domain features, which may limit its applicability in certain situations. For example, for data with unclear or difficult to extract frequency domain features, the performance of FreDF may not be as expected.

**Questions:**

1. Computational complexity of frequency domain conversion: Will the frequency domain conversion mentioned in the paper significantly increase computational complexity? How efficient is the FreDF method when dealing with large-scale datasets?
2. Generalization ability: How does the FreDF method generalize across different domains and datasets? Has it been tested and validated in more practical application scenarios?
3. Applicability of frequency domain features: How effective is the FreDF method for data with unclear or difficult to extract frequency domain features? Are there any relevant experimental results or analysis?
4. Model interpretability: Does the FreDF method have interpretability in the frequency domain? How do users understand and interpret these predicted results?

---

> ### Author Response · Authors · 2024-11-22
>
> Thank you very much for your positive comments and appreciation of our **novelty, generality and empirical performance**. Below are our responses to the specific concerns and queries.
>
> We observed that some comments in the weaknesses and questions sections convey similar points. We will selectively merge them to make the response concise and easy to follow.
>
> ---
>
> #### **[Q1,W1] Computational complexity of frequency domain conversion: Will the frequency domain conversion mentioned in the paper significantly increase computational complexity? How efficient is the FreDF method when dealing with large-scale datasets?**
> **Response.** Thank you for your insightful question. We acknowledge the importance of investigating the complexity of FreDF, typically the FFT for domain conversion. Since FFT is executed for each label sequence, it is the label sequence length, denoted as $T$, that mainly impacts the executive efficiency. Therefore, our discussion focuses on the complexity of FFT for varying values of $\mathrm{T}$.
> - **In the training stage, FFT operates with a computational complexity of $\mathcal{O}(\mathrm{T}\log \mathrm{T})$, ensuring scalability through its logarithmic scaling.** To validate this, we measured FFT processing times across various sequence lengths. As shown in Table 1, the FFT operation introduces minimal overhead, with the maximum observed time being **0.076 ms for $T=720$**. Therefore, FFT does not increase the computational complexity significantly of model training.
>
> |Stage| Time|
> |---|---|
> |T=96	|0.0685±0.0059|
> |T=192	|0.0686±0.0076|
> |T=336	|0.0708±0.0143|
> |T=720	|0.0760±0.0241|
> - **In the inference stage, the frequency loss computation is not required**, eliminating any additional computational overhead associated with FFT. Consequently, the FFT does not increase any computational complexity of model inference.
> - In conclusion, the complexity of FreDF is ignorable in both training and inference stages. It ensures that FreDF remains scalable for long label sequence and large datasets. **We have added these discussions in the revised Appendix E.2.**
>
> #### **[Q2,W2] Generalization ability: How does the FreDF method generalize across different domains and datasets? Has it been tested and validated in more practical application scenarios?**
> **Response.** We appreciate the reviewer’s emphasis on the generalization capabilities of FreDF. To validate FreDF across diverse application domains, we undertook the following efforts.
> - **Firstly, we tested FreDF on nine datasets spanning various domains.** The consistent performance of FreDF across these datasets demonstrates its applicability to different real-world scenarios.
>
> |Domain| Dataset|
> |---|---|
> |Health|ETTh1, ETTh2, ETTm1, ETTm2|
> |Electricity|ECL|
> |Transportation|Traffic, PEMS03, PEMS08|
> |Meteorology	|Weather|
> |**Process engineering**| **SRU** |
>
> - **Secondly, to further validate FreDF’s generality, we have added experiments on the SRU dataset from the chemical process engineering domain.** This dataset comprises monitoring logs from a Sulfur Recovery Unit, featuring complex correlations among multiple sensor readings over time. FreDF significantly outperformed baseline methods on this challenging dataset, underscoring its effectiveness in practical, high-stakes environments.
>
> | Metrics | FreDF (Ours) || iTransformer || FreTS || TimesNet || MICN || TiDE || DLinear || FEDformer || Autoformer || Transformer ||
> |---------|------|--------|-------------|--------|-------|-------|------|-----------|---------|----|-------|------|-------|---------|-------|------------|-----------|--------|---------|----------|
> |Horizons| MSE | MAE    | MSE | MAE | MSE | MAE    | MSE | MAE         | MSE | MAE         | MSE | MAE | MSE | MAE | MSE | MAE | MSE | MAE | MSE | MAE  |
> | 96      | **0.666** | **0.434** | 0.722 | 0.455    | 0.701 | 0.507 | 0.686 | 0.459 | 0.771 | 0.557 | 0.711 | 0.455 | 0.764 | 0.555 | 0.783 | 0.549 | 0.879 | 0.607 | 1.540 | 0.895 |
> | 192     | **0.951** | **0.558** | 0.987 | 0.571    | 1.025 | 0.660 | 0.958 | 0.569 | 1.084 | 0.693 | 0.982 | 0.572 | 1.054 | 0.672 | 1.084 | 0.662 | 1.050 | 0.629 | 2.096 | 1.037 |
> | 336     | **1.325** | **0.687** | 1.381 | 0.707    | 1.460 | 0.808 | 1.332 | 0.696 | 1.464 | 0.812 | 1.366 | 0.704 | 1.446 | 0.808 | 1.402 | 0.732 | 1.406 | 0.733 | 2.528 | 1.137 |
> | 720     | **1.641** | **0.840** | 1.655 | 0.849    | 1.747 | 0.959 | 1.621 | 0.848 | 1.512 | 0.876 | 1.660 | 0.850 | 1.555 | 0.897 | 1.618 | 0.851 | 1.665 | 0.867 | 2.537 | 1.198 |
> | **Avg**      | **1.146** | **0.630** | 1.186 | 0.645    | 1.233 | 0.734 | 1.150 | 0.643 | 1.208 | 0.735 | 1.180 | 0.645 | 1.205 | 0.733 | 1.222 | 0.698 | 1.250 | 0.709 | 2.175 | 1.067 |

---

> ### Author Response · Authors · 2024-11-22
>
> #### **[Q3,W3] Applicability of frequency domain features: How effective is the FreDF method for data with unclear frequency domain features? Are there any relevant experimental results or analysis?**
> **Response.** Thank you for your meticulous comment regarding the effectiveness of FreDF in cases where frequency features are unclear. **Our analysis and experiments demonstrate that FreDF maintains effective even under these conditions.**
> - FreDF is fundamentally designed to mitigate label correlation by utilizing the FFT as a mathematical tool. This approach is agnostic to the clarity of inherent frequency features within the data, as demonstrated in Theorem 3.3. Consequently, FreDF effectively improves forecast performance by reducing label correlations regardless of whether the frequency components are clear.
> - To empirically evaluate FreDF’s efficacy under these conditions, we introduced a variant named FreDF-D, which applies FFT along the variable dimension instead of the time dimension. The frequency features are unclear when performing FFT along the variable dimension. The results are shown in the table below.
>
> | Data           | ETTh1|||       | ETTm1|||        | ECL |||          |
> |------------------|------------|------------|------------|------------|------------|------------|---|---|---|---|---|---|
> | Metric                | MSE        | ΔMSE           | MAE        | ΔMAE           | MSE        | Δ MSE          | MAE        | Δ MAE          | MSE        | Δ MSE          | MAE        | Δ MAE          |
> | iTransformer     | 0.449      | -           | 0.447      | -           | 0.415      | -           | 0.416      | -           | 0.176      | -           | 0.267      | -           |
> | + FreDF        | 0.437      | $\downarrow$2.63%     | 0.435      | $\downarrow$2.62%     | 0.392      | $\downarrow$5.49%     | 0.399      | $\downarrow$4.01%     | 0.170      | $\downarrow$3.41%     | 0.259      | $\downarrow$2.77%     |
> | + FreDF-D        | 0.445      | $\downarrow$0.92%     | 0.440      | $\downarrow$1.42%     | 0.395      | $\downarrow$4.77%     | 0.398      | $\downarrow$4.33%     | 0.171      | $\downarrow$2.51%     | 0.260      | $\downarrow$2.52%     |
>
> - The results indicate that FreDF-D bring comparable performance gain with FreDF, demonstrating FreDF’s capability to handle label correlations even when frequency features are not explicitly clear. By the way, FreDF slightly outperforms FreDF-D, underscoring the importance of temporal auto-correlation in label sequences, which is exactly the focus of this paper.
>
>
>
> #### **[Q4] Model interpretability: Does the FreDF method have interpretability in the frequency domain? How do users understand and interpret these predicted results?**
> **Response.** Thank you for your insightful comment on interpretability. Yes, FreDF offers enhanced interpretability compared to traditional DF with temporal losses. We are pleased to communicate the key points with you as follows.
>
> - **Interpretability of labels and predictions. Transforming time series data into the frequency domain facilitates a clearer understanding of complex patterns.** For instance:
>   - A Linear Frequency Modulation (LFM) signal in the frequency domain appears as a linear function, explicitly illustrating the modulation process over time. This clear representation aids in comprehending the underlying signal dynamics.
>   - Consider respiratory sounds from healthy individuals versus patients with pneumonia. In the time domain, both may appear as noisy sequences, making differentiation challenging. However, in the frequency domain, pneumonia-induced respiratory sounds exhibit increased energy at specific frequencies, enabling easier identification and interpretation of pathological conditions.
>
>
> - **Interpretability of loss components. Analyzing forecast errors across different frequency components provides valuable insights into the model’s learning process and guides the refinement of training strategies.** For instance:
>   - If the model exhibits high errors in low-frequency components, it indicates that the model fails to learn basic, slowly changing semantics in the dataset. In this case, we could filter out high-frequency noise, or increase the loss weight for low-frequency components, to prioritize the learning of low-frequency components.
>   - If the model exhibits high errors in high-frequency components, while low-frequency components are accurately predicted, it suggests that the model has captured the fundamental semantics but struggles with finer details. In this case, we could increase the loss weight for high-frequency components to encourage the model to learn more detailed patterns.
>
> - **In conclusion, FreDF indeed has advantage of interpretability which enable users to understand dataset property and diagnose model performance.** Nevertheless, in this work we contemporally focus on its efficacy to improve model performance by diminishing the bias from label correlation.

---

> ### Author Response · Authors · 2024-11-30
>
> Dear Reviewer  3nNT,
>
> We sincerely appreciate the time and effort you have dedicated to reviewing our manuscript. We are delighted that you recognize the significance of our work in identifying and eliminating bias, find our method novel and interesting, and consider our empirical results promising. In response to your valuable feedback, we have undertaken the following actions:
>
> |Comment|Answers and Actions|
> |---|---|
> |W1&Q1: Complexity| We have discussed the computational complexity of FreDF and empirically demonstrated its efficiency. Specifically, the maximum observed processing time is 0.076 ms for the largest forecast horizon  T=720 . These details are involved in Appendix E.2.
> |W2&Q2: Generalization in different domain datasets| FreDF was tested on nine datasets spanning various domains, including healthcare, electricity, transportation, and meteorology. To further validate its generality, we have included additional experiments on the SRU dataset from the chemical process engineering domain. These results reinforce FreDF’s applicability across diverse real-world scenarios.
> |W3&Q3: Frequency features | We introduced a variant named FreDF-D, which applies FFT along the variable dimension instead of the time dimension, resulting in less clear frequency features. The performance of FreDF-D is comparable to that of FreDF, demonstrating FreDF’s efficacy in handling label correlations and eliminating bias even when frequency features are not clear.
> |Q4: Interpretability | We discussed the potential of FreDF in terms of interpretability from two aspects, which enable users to understand dataset property and diagnose model performance.|
>
> As the discussion draws to a close, we are wondering whether our responses have properly addressed your concerns? Your feedback would be extremely helpful to us. If you have further comments or questions, we hope for the opportunity to respond to them.
>
> Many thanks,
>
> 13602 Authors

---

### Official Review · Reviewer_ErxY · 2024-11-03

**Soundness:** 4
**Presentation:** 3
**Contribution:** 3
**Rating:** 8
**Confidence:** 4

**Summary:**

This work introduces a new loss term for multi-step time-series forecasting that penalizes errors within a decorrelated representation space of the ground truth labels. In its current formulation, this decorrelated space is defined as the frequency representation of both labels and forecasts. Experimental results demonstrate that the proposed approach significantly enhances forecasting accuracy across various datasets and base models.

**Strengths:**

1. The authors identify and address the bias introduced by label correlation in time-series modeling, which is a novel issue for me and holds substantial potential generality across different scenarios.
2. The method is sound, straightforward and shows very promising results. The theoretical results are relatively persuasive, demonstrating the bias caused by label correlation, and subsequently FreDF's elimination of label correlation and thereby bias.
3. The experiment is comprehensive. Extensive set of experiments showed that: the approach contributes to the state-of-the-art, different components of the loss contribute to performance, the approach is robust to hyperparameter values.

**Weaknesses:**

1. It seems that this work aims to penalize errors in a space that decorrelates labels. Experiments mainly achieve de-correlation via FFT or FFT2. It will be beneficial to explore the efficacy of other transformations beyond the Fourier transform?
2. While the introduction discusses various forecasting models, it could benefit from a stronger focus on established forecasting paradigms (e.g., direct forecasting, iterative forecasting). The inclusion of forecast models (iTransformer, Linear) may detract from highlighting the contribution and role of FreDF that seems to be orthogonal to forecast models.
3. The source code should be refined. The current implementation comprises numerous scripts, and it is somewhat unclear how each script relates to the experiments discussed in the manuscript. Besides, the current environment setup appears to depend on unexpected repositories like `torchmetrics` and `patool`. Including a `Dockerfile` or `conda` environment file would help a lot.

**Questions:**

1. In the experiments, how to determine the hyperparameters of FreDF and baselines?
2. According to the ablation study, the frequency term seems to be almost all that's needed. Is it correct?
3. During my attempt to reproduce the experiments, I encountered an error (`AttributeError: 'Exp_Short_Term_Forecast' object has no attribute 'seasonal_patterns'`) when running the short-term forecasting experiments. Is this error expected, or does it indicate a potential issue in the codebase?

Overall I think this work is interesting and promising. I will consider raising my score if my questions could receive positive responses, especially my major concerns (W1, Q1, Q3).

---

> ### Author Response · Authors · 2024-11-22
>
> Thank you very much for your encouraging support and appreciation of our **novelty, presentation, methodology and empirical results**. Below are our responses to the specific query raised.
>
> ---
>
> #### **[W1] It seems that this work aims to penalize errors in a space that decorrelates labels.**
> **Response.** Thank you for your insightful question. We agree that exploring other transformations can enhance the technical quality of this manuscript.
> - To investigate the generality with different transformations, motivated by the fact that FFT can be viewed as projections onto Fourier polynomials, **we extended the implementation of FreDF by replacing FFT with projections onto other established polynomials (Legendre, Chebyshev, Laguerre)**. Each of these polynomials is adept at capturing specific temporal patterns such as trends (Legendre polynomial) and periodicity (Fourier polynomial).
> - **The results are summarized in Figure 5, where we observed that Legendre and Fourier bases demonstrate superior performance**. This superiority is attributed to the orthogonality between polynomials, a feature not guaranteed by others, as analyzed in Appendix C. It underscores the importance of orthogonality when selecting polynomials for implementing FreDF.
>
> #### **[W2] While the introduction discusses various forecasting models, it could benefit from a stronger focus on established forecasting paradigms (e.g., direct forecasting, iterative forecasting).**
> **Response.** Thank you for your valuable advice. We agree that forecasting models are orthogonal to the contribution of FreDF. Nevertheless, we believe that including a discussion of these models in the introduction is crucial for two reasons. Firstly, the development of forecast models (iTransformer, DLinear, etc.) is indeed a recent focus in time series modeling. Introducing them could engage readers in this field and contextualize our work. Secondly, after introducing forecast models, we can directly clarify the distinct contribution of FreDF in addressing label correlation, independent of the input correlation managed by forecast models.
>
>
>
> #### **[W3] The source code should be refined.**
> **Response.** We are pleased that you are willing to read our code and reproduce our results.
> - Firstly, we clarify that all experimental scripts are located in `./scripts/`. Typically, the main results of FreDF can be reproduced by executing the bash script in `./scripts/fredf_exp`. For example, you can reproduce the long-term forecast, short-term forecast and imputation results by executing:
>   > \# long-term forecast
>
>   > bash ./scripts/fredf_exp/ltf_overall/ETTh1_script/iTransformer.sh
>
>   > \# short-term forecast
>
>   > bash ./scripts/fredf_exp/stf_overall/FreTS_M4.sh
>
>   > \# imputation
>
>   > bash ./scripts/fredf_exp/imp_autoencoder/ETTh1_script/iTransformer.sh
> - Secondly, we agree that a file for constructing environments could be important for reproduction. We have uploaded a `requirement.txt` file in the updated repository, which encapsulates all packages and dependencies in our environment.
> - **Action. We have updated the README.md file to include these examples, which should help users understand how to navigate the code. Furthermore, we have uploaded a requirements.txt file for environment configuration. We hope these efforts will assist you in reproducing our results.**
>
>
> #### **[Q1] In the experiments, how to determine the hyperparameters of FreDF and baselines?**
> **Response.** Thank you very much for your kind query. The implementation of FreDF is built upon the repository Time Series Library (TSLib). Therefore, in our experiments, we adhere to the hyperparameters suggested by TSLib for the baseline methods. For FreDF, we fix the hyperparameters associated with the forecast models, such as the iTransformer and TimesNet, and only fine-tune the frequency loss strength $\alpha$ and the learning rate conservatively on the validation set. This approach allows us to isolate the contribution of FreDF without introducing gains from tuning the forecast models, thereby making the empirical advantages of FreDF persuasive.

---

> ### Author Response · Authors · 2024-11-22
>
> #### **[Q2] According to the ablation study, the frequency term seems to be almost all that's needed. Is it correct?**
> **Response.**  Thank you for your meticulous observation. Your point is valid, and we have also observed that the fusing loss $\mathcal{L}^\alpha$ produces marginal improvements over $\mathcal{L}^\mathrm{feq}$.
> - This phenomenon, in our view, reinforces the strength of FreDF: the ability to achieve most performance gains, without intricate tuning of $\alpha$ (directly setting $\alpha=1$), makes FreDF easy to use in practice.
> - The role of the temporal loss is two-fold. Firstly, the improvement from $\mathcal{L}^\mathrm{feq}$ to $\mathcal{L}^\alpha$ is sometimes not ignorable. For instance, an MAE improvement of 0.003 is observed on Weather, which is comparable to the gap between TiDE (2023) and iTransformer (2024). Secondly, the parameter $\alpha$ allows for adjusting the contribution from each loss flexibly, which makes it feasible to thoroughly evaluate the efficacy of the frequency loss by analyzing the forecast performance across different $\alpha$ values. The results show that as we incorporate $\alpha$ and moderately increase it, the forecast performance improves, which convincingly supports the efficacy of the proposed frequency loss.
>
> #### **[Q3] During my attempt to reproduce the experiments, I encountered an error.**
> **Response.**  We sincerely apologize for this oversight. This error arises because we did not define `self.seasonal_patterns` before `_build_model`. We have corrected this issue in the updated repository. Additionally, we have conducted comprehensive tests on multiple devices to ensure the robustness of the revised codebase. We appreciate the reviewer for identifying this problem and regret any inconvenience it may have caused.

---

> > ### Comment · Reviewer_ErxY · 2024-11-23
> >
> > Your point is valid and I have produced the experimental results using the updated materials (But I do not verify all experiments.)
> >
> > I also appreciate the current efforts that explore the generality of FreDF  by using different polynomials. For your information, what I think would be fruitful for further investigatation are alternative transform paradigms, such as ICA. Omitting this does not compromise the integrity of the current version, but it would be interesting to explore it.
> >
> > Since my primary concerns have been addressed, I am pleased to adjust my rating accordingly.

---

> ### Author Response · Authors · 2024-11-24
> **Thank you for your continued and increased support!**
>
> Dear ErxY,
>
> We are pleased that our responses address your concerns and help your verification of our empirical results.
>
> Your clarified suggestion is valuable and we promise conducting the corresponding experiments.
>
> Big thanks!
>
> 13062 Authors

---

### Author Response · Authors · 2024-11-25

We are grateful to all reviewers for their helpful comments regarding our paper. We are pleased that Reviewers **ErxY, 3nNT, and WM7u** recognize the **significance of our identification and elimination of bias, find our method novel and interesting, and consider our empirical results promising**. Below is a summary of our responses:
- To Reviewer ErxY: We have updated the repository to facilitate ease of reproduction, and clarified the hyperparameter selection process.
- To Reviewer 3nNT: We have emphasized the efficiency advantages of FreDF, added new experiments using an additional dataset, and discussed the potential of FreDF regarding interpretability.
- To Reviewer WM7u: We have included experiments across different random seeds, provided an extended version of Theorem 1, corrected typographical errors.
- To Reviewer 6T6T: We have  thoroughly investigated, discussed, and compared the listed references.  Additionally, we conducted experiments to reproduce and compare **ALL** listed references over six working days, resulting in a comprehensive response. **The conclusion is that FreDF is differentiated from these works in multiple aspects and makes unique contributions, particularly in the identification and elimination of bias caused by label correlation.**

Please refer to our detailed responses to each point raised. We have also uploaded a revised manuscript with changes highlighted in blue. We hope that our revisions and clarifications satisfactorily address the concerns raised.
Thank you again for your valuable time and expertise.

---

### Meta-Review · Area_Chair_KEAi · 2024-12-18

**Metareview:**

The paper considers the problem of learning in time series in settings with label autocorrelation. The authors identify a gap in how existing methods behave in these settings. They then propose a new method that can address this gap. The paper includes theoretical analyses and validates the method on synthetic and real-world datasets.

Overall, the feedback from reviewers was positive. On the one hand, the paper was praised for its innovative focus on label autocorrelation and rigorous theoretical analysis. Reviewers highlighted its potential impact on improving time series modeling and its strong experimental results. On the other hand, the paper lacked specific motivating examples and a discussion of practical applications where label autocorrelation is relevant. Having read the reviews, the rebuttal, and the paper, I am recommending that the paper be accepted at this time.

In addition to the changes requested by reviewers, I recommend that the authors make two simple changes that can improve the impact:

1. **Motivating Applications**: The paper is missing some discussion of when/where we could "inherent autocorrelation in the label sequence."

- Abstract and Intro: These sections include claims like: "Time series modeling is uniquely challenged by the presence of autocorrelation in both historical and label sequences." Both sections would be far more compelling if you could add an example of where labels exhibit autocorrelation, and some concrete detail of what can go wrong.

- Body: The paper could use a more detailed discussion. In general, readers should know the kinds of applications where they can plausibly expect label autocorrection, or where they can ignore this possibility. Listing potential tests could help as well.

2. **Rephrase/Change Title:** I would strongly advise considering updating the title of your paper. The current title has two problems: (a) it's confusing since it reads as a list of nouns ("biases" could be read as a noun too); (b) it does not clearly describe your contribution (there are many different ways to "bias" a forecast; also, it doesn't make it clear if you are developing a method or studying the problem). Some alternatives include: "Learning [Time Series] with Label Autocorrelation", or "Learning Time Series in the Frequency Domain."

**Additional Comments On Reviewer Discussion:**

See above

---

### Decision · Program_Chairs · 2025-01-22

Accept (Poster)

---

> ### Public Comment · ~Hao_Wang28 · 2025-03-15
> **Updates in the Camera-Ready Version.**
>
> We express our sincere appreciation to the helpful comments from reviewers and the area chair.  Currently, the camera-ready of this work has been available. In preparing the final camera-ready version, we have meticulously incorporated all the changes suggested during the rebuttal phase and have undertaken additional revisions to enhance the presentation of our manuscript.
> - In alignment with the insightful suggestion from the area chair in the metareview, we have revised the title to "FreDF: Learning to Forecast in the Frequency Domain" to better reflect the core contribution of our work.
> - We have carefully addressed typographical errors and refined the narrative to improve readability. Additionally, we have adjusted the overall layout of the paper to ensure a more coherent and logical flow of ideas.
> - Recognizing that a lengthy appendix attached to the main text might hinder readers from efficiently locating relevant information, we have reorganized the appendix into two distinct parts. The components deemed less critical for core understanding (but necessary for comprehensiveness) of this work have been moved to the supplementary appendix, to make the main text focused.
>
> Once again, we sincerely thank the reviewers and the area chair for their comments and guidance, which has been instrumental in shaping this work.